# Wnt/β-catenin signaling is an evolutionarily conserved determinant of chordate dorsal organizer

Iryna Kozmikova*, Zbynek Kozmik

Laboratory of Transcriptional Regulation, Institute of Molecular Genetics of the Czech Academy of Sciences, Prague, Czech Republic

**Abstract** Deciphering the mechanisms of axis formation in amphioxus is a key step to understanding the evolution of chordate body plan. The current view is that Nodal signaling is the only factor promoting the dorsal axis specification in the amphioxus, whereas Wnt/β-catenin signaling plays no role in this process. Here, we re-examined the role of Wnt/βcatenin signaling in the dorsal/ventral patterning of amphioxus embryo. We demonstrated that the spatial activity of Wnt/β-catenin signaling is located in presumptive dorsal cells from cleavage to gastrula stage, and provided functional evidence that Wnt/β-catenin signaling is necessary for the specification of dorsal cell fate in a stage-dependent manner. Microinjection of *Wnt8* and *Wnt11* mRNA induced ectopic dorsal axis in neurulae and larvae. Finally, we demonstrated that Nodal and Wnt/β-catenin signaling cooperate to promote the dorsal-specific gene expression in amphioxus gastrula. Our study reveals high evolutionary conservation of dorsal organizer formation in the chordate lineage.

## Introduction

One of the most intriguing queries in developmental biology is how the body plan evolved throughout the animal kingdom. In vertebrates, one of the key initial events in the early development is the formation of dorsal (Spemann) organizer, which establishes early dorsoanterior/ventroposterior asymmetry. The dorsal organizer was first described in amphibian embryo at the early gastrula stage as a group of cells from dorsal blastopore lip, which has the ability to induce the formation of secondary embryonic axis (a tween embryo) if grafted to the ventral side of the host embryo (*Spemann and Mangold, 1924*). It was subsequently shown that dorsal organizer functions in all phyla of vertebrates including fish (*Oppenheimer, 1936*), birds (*Waddington and Schmidt, 1933*) and mouse (*Beddington, 1994*), suggesting that conserved mechanisms of body plan formation are present in all vertebrates. Moreover, it was demonstrated that the dorsal organizer, which is located in the dorsal blastopore lip of gastrula embryo, is also present in the cephalochordates (*Le Petillon et al., 2017*; *Tung and Yeh, 1961*), which represent an early branch of the chordate group and share with vertebrates the typical chordate features such as notochord, dorsal hollow neural tube, pharyngeal slits, and post-anal extending tail.

In vertebrates, one of the prerequisites during dorsal organizer formation is accumulation of nuclear β-catenin in the dorsal territory of early blastula embryo (*Schneider et al., 1996*; *Roeser et al., 1999*; *Schohl and Fagotto, 2002*), which is critical for the initial specification of the dorsal cell fate (*Kelly et al., 2000*; *Bellipanni et al., 2006*; *Heasman et al., 1994*). The role of Wnt/β-catenin signaling in the dorsal/ventral (DV) axis establishment was first demonstrated in *Xenopus* by gain-of-function experiments, in which ectopic Wnt proteins induced secondary organizer formation and duplication of the embryonic axis (*McMahon and Moon, 1989*; *Smith and Harland, 1991*; *Sokol et al., 1991*). Maternal Wnt/β-catenin signaling induces expression of genes encoding transcription factors and secreting proteins that are involved in the initial formation of Spemann

*For correspondence:
kozmikova@img.cas.cz

**Competing interests:** The authors declare that no competing interests exist.

organizer in the *Xenopus* embryo (*Tao et al., 2005*). These early activated targets include genes encoding the Nodal-related group of proteins (*Kelly et al., 2000*; *McKendry et al., 1997*; *Yang et al., 2002*; *Takahashi et al., 2000*; *Ding et al., 2017*). Numerous studies have demonstrated that Nodal signaling is required for the dorsal mesoderm formation and establishment of organizer in *Xenopus* (*Takahashi et al., 2000*; *Jones et al., 1995*; *Osada and Wright, 1999*; *Birsoy et al., 2006*; *Agius et al., 2000*; *Hoodless et al., 1999*), zebrafish (*Feldman et al., 1998*) and mammals (*Niederländer et al., 2001*; *Martyn et al., 2018*; *Gritsman et al., 2000*; *Zhou et al., 1993*; *Collignon et al., 1996*; *Chea et al., 2005*). The current view is that maternal Wnt/β-catenin signaling and Nodal signaling induce a variety of transcription factors and secreting proteins that act during the cleavage and blastula stages to establish a low level of Bmp signaling activity mediated by phosphorylated Smad (P-Smad) transcription factors P-Smad1, P-Smad5, or P-Smad8 and a high level of Nodal/P-Smad2-mediated signaling activity in the dorsal territory at the late blastula and gastrula stages. In contrast, a low level of Nodal/P-Smad2-mediated signaling activity and a high level of Bmp/P-Smad1/P-Smad5/P-Smad8-mediated signaling activity is established in the ventral territory of the embryo. The establishment of these two opposing gradients is crucial for proper specification of the dorsal and ventral cell fate and axial patterning of the embryo in vertebrates (*Takahashi et al., 2000*; *Jones et al., 1995*; *Osada and Wright, 1999*; *Lee et al., 2001*; *Piccolo et al., 1996*; *Xanthos et al., 2002*; *Xu et al., 2014*) and in cephalochordate amphioxus (*Le Petillon et al., 2017*; *Onai et al., 2010*; *Morov et al., 2016*; *Kozmikova et al., 2013* and reviewed in *Zinski et al., 2018*). Noteworthy, the opposing gradients of Nodal/P-Smad2-mediated signaling activity and P-Smad1/5/8-mediated signaling activity, although promoted by BMP-like ligands ADMP1 and ADMP2, operate to establish the DV embryonic axis in a representative of echinoderms closely related to chordates (*Lapraz et al., 2015*; *Saudemont et al., 2010*; *Lapraz et al., 2009*). Similarly as in vertebrates, in sea urchin Wnt/β-catenin is required for initiation of *Nodal* expression at the blastula stage (*Yaguchi et al., 2008*; *Duboc et al., 2004*; *Range et al., 2007*) in the region that is suggested to be the functional equivalent of Spemann organizer (*Lapraz et al., 2015*). This suggests deep evolutionary conservation of the molecular mechanisms operating during axial patterning of deuterostome. However, in cephalochordate amphioxus the role of Wnt/β-catenin signaling in dorsoanterior/ventroposterior patterning and dorsal organizer formation is not clear. A recent study suggests that Wnt/β-catenin signaling functions in animal/vegetal axial patterning and early mesoderm specification in the amphioxus embryo, although there is no data showing asymmetrical distribution of β-catenin (*Onai, 2019*). Previous observations of nuclear distribution of β-catenin during the cleavage and early gastrula stage in two different amphioxus species *Branchiostoma belcheri* and *Branchiostoma floridae* are contradictory (*Holland et al., 2005*; *Yasui et al., 2002*). One study suggested that the concentration of nuclear β-catenin is higher on the dorsal side of the embryo at the onset of gastrulation (*Yasui et al., 2002*). The other study showed that nuclear β-catenin is mainly localized in the region of future ectoderm at the onset of gastrulation and throughout the ectoderm at mid-gastrula stage (*Holland et al., 2005*). In these studies, researchers addressed the role of Wnt/β-catenin in dorsoanterior/ventroposterir patterning by using the treatment with high concentration of LiCl, an inhibitor of Gsk3β, which activates canonical Wnt/β-catenin signaling. After the treatment of the embryos with LiCl from the one-cell to 128 cell stage, Yasui et al. observed that Goosecoid, Otx and Lhx3 expression domains changed from asymmetrical dorsal to radially symmetric pattern at the initial gastrulation stage (*Yasui et al., 2002*). However, the authors argue that the embryos were able to establish DV asymmetry because the expression domains of Pitx and FoxA were unaffected. Holland et al. did not observe any effect of treatment with LiCl on the establishment of dorsal cell fate in the embryos treated during the blastula stage, and therefore concluded that the involvement of Wnt/β-catenin signaling in the establishment of DV polarity is a vertebrate innovation (*Holland et al., 2005*). If so, then taking into consideration the research on sea urchin (*Lapraz et al., 2015*; *Yaguchi et al., 2008*) one can postulate that during evolution, the molecular mechanism involving Wnt/β-catenin in the establishment of dorsal organizer and dorsoanterior/ventroposterior axis was lost in basal chordate amphioxus, and plausibly, in the common ancestor of chordates. To date, it has been suggested that only Nodal signaling itself is required for the dorsal cell fate specification in the early amphioxus embryo, despite the fact that it is still unknown where the Nodal signaling is active during first steps of amphioxus development (*Le Petillon et al., 2017*; *Onai et al., 2010*; *Morov et al., 2016*).

Therefore, in our study, we reinvestigated the activity of Wnt/β-catenin signaling by detecting the nuclear β-catenin distribution with specific anti-amphioxus β-catenin antibodies. Further, in the gain-of-function and loss-of-function experiments, we reexamined whether Wnt/β-catenin signaling plays a role in the formation of dorsal organizer in the amphioxus. Additionally, we described how the mediator of Nodal signaling P-Smad2 (phosphorylated Smad2) is distributed in early amphioxus embryo.

## Results

### Asymmetrical distribution of nuclear β-catenin at the cleavage, blastula and early gastrula stages of amphioxus embryo

In previous studies, which explored the activity of Wnt/β-catenin signaling during early amphioxus development and obtained different results, the pattern of nuclear β-catenin was detected with distinct antibodies specific for human and chicken β-catenin (*Holland et al., 2005*; *Yasui et al., 2002*). To reinvestigate the distribution of nuclear β-catenin during amphioxus early development, we first performed immunostaining of the embryos of *Branchiostoma lanceolatum* at the cleavage, blastula and early gastrula stages by using homemade mouse antibodies specific for amphioxus β-catenin. As previously demonstrated, these antibodies detect nuclear β-catenin in the amphioxus embryos in the same pattern as commercial rabbit anti-human β-catenin antibodies (*Bozzo et al., 2017*). Imaging of the stained embryos was done with a confocal microscope by recording z-stacks (optical sections) of the whole amphioxus embryo, which allowed us to examine every individual nucleus and cell, and subsequently perform three-dimensional (3D) images. In parallel, by in situ hybridization we detected expression of the *Axin* gene, which is a target of Wnt/β-catenin signaling in the amphioxus (*Kozmikova et al., 2011*), and therefore may serve as a readout of Wnt/β-catenin signaling. We did not observe nuclear β-catenin at the initial cleavage stages (*Figure 1A–A'* and *Figure 1—figure supplement 1A*-Aiii, B-Biv and C-Civ). Nuclear β-catenin was first detected at the 32-cell stage; its distribution was asymmetrical with higher signal in the vegetal pole of the embryo (*Figure 1B–C'*). At the 64-cell stage, nuclear β-catenin was concentrated in the cells of vegetal hemisphere (*Figure 1D–D'*). Similarly, the expression of amphioxus *Axin* was stronger in one domain of the embryo (*Figure 1E*). The asymmetrical pattern of nuclear β-catenin distribution and *Axin* expression was preserved at the blastula stage (*Figure 1F–F'* and *Figure 1G*, respectively). At the initial gastrulation stage, nuclear β-catenin and stronger *Axin* expression was located in one half of the flattened vegetal hemisphere (*Figure 1H–H'* and *Figure 1I*). In *Branchiostoma lanceolatum*, the dorsal endomesoderm region of blastopore is morphologically distinguishable from the ventral endomesoderm at the early gastrula stage (*Figure 1J*). As compared to the ventral endomesoderm, the cells of the dorsal endomesoderm are more elongated and closely attached to one another (*Figure 1J–L*). In these cells, nuclear β-catenin (*Figure 1J'*) detected using rabbit anti-human β-catenin antibodies was co-expressed with dorsal-specific transcription factor Goosecoid (*Figure 1J''*). We confirmed the presence of nuclear β-catenin in the dorsal endomesoderm using mouse anti-amphioxus β-catenin antibodies (*Figure 1K–K'*). Furthermore, a strong expression of *Axin* was detected in this dorsal domain (*Figure 1L*). At the cap-shaped gastrula stage the expression of both nuclear β-catenin and *Axin* was higher in the dorsal region of the embryo (*Figure 1M–N*). At the late gastrula stage, the signal remained stronger dorsally, but was more concentrated posteriorly (*Figure 1—figure supplement 1D-Dv*). At the early neurula stage, nuclear β-catenin was detected only in the posterior domain of the embryo (*Figure 1—figure supplement 1E-Ev*). These data demonstrate that the activity of Wnt/β-catenin signaling is dynamic during the early development of amphioxus embryo and nuclear β-catenin is asymmetrically distributed during the cleavage and gastrula stages.

### Nuclear β-catenin is localized in the same region as organizer-specific gene *Goosecoid* and mediator of nodal signaling P-Smad2 during early amphioxus development

To comprehensively characterize where the Wnt/β-catenin signaling is active during early *Branchiostoma lanceolatum* development, we performed double immunostaining with mouse anti-amphioxus Goosecoid and rabbit anti-human β-catenin antibodies at the cleavage, blastula and early gastrula stages. The expression of Goosecoid protein was first detected at the 256-cell stage (*Figure 2A–Ai*

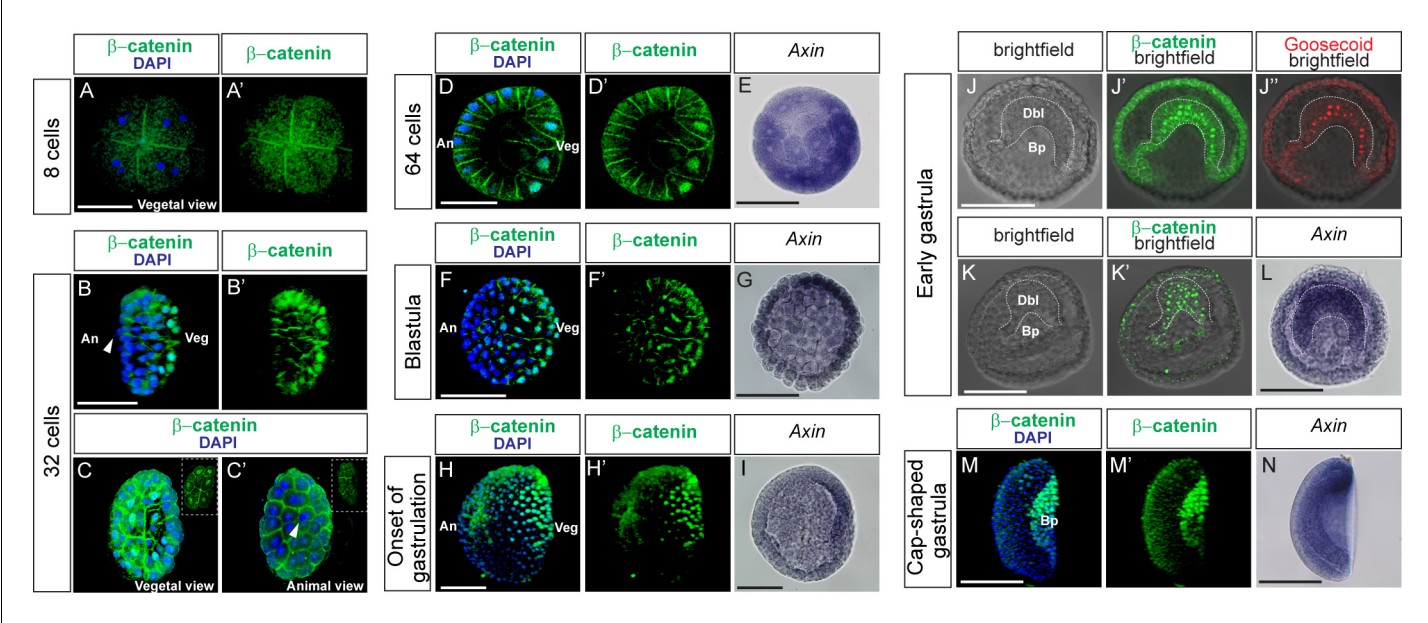

**Figure 1.** Asymmetric activity of the Wnt/β-catenin signaling pathway during the cleavage and early gastrula stages of amphioxus embryo. (A–D'), (F–F'), (H–H'), (K') and (M–M') Immunostaining of β-catenin with specific anti-amphioxus β-catenin antibody produced in the mouse. (J) Bright-field of the embryo shown with the signals in (J') and (J''). (J') Immunostaining with anti-human β-catenin antibody produced in the rabbit. (J'') Immunostaining of the same embryo as in (J') with specific anti-amphioxus Goosecoid antibody produced in the mouse. (K) Bright-field of the embryo shown with the signals in (K'). (E), (G), (I), (L) and (N) In-situ hybridization of the Wnt/β-catenin signaling target gene *Axin*. (A–A') β-catenin is not present in the nuclei of the embryos at 8-cell stage. Embryos are shown from the vegetal view. (B–C') Nuclear β-catenin is accumulated in the vegetal half of the embryo at 32-cell stage. Arrowheads show polar body, which demarcates the animal pole. (B–B') side view, (C) vegetal view, (C') animal view. Inserts in (C) and (C') demonstrate individual optical sections from the vegetal and animal poles, respectively. (D–D') Individual optical section demonstrates that β-catenin is present in the vegetal nuclei of a 64-cell stage embryo. (E) The expression of the Wnt/β-catenin signaling target gene *Axin* is not ubiquitous in the 64-cell stage embryo. (F–F') Asymmetrical distribution of β-catenin and asymmetrical expression of *Axin* (G) in the blastula of amphioxus embryo. (H–H') Nuclear β-catenin and (I) the expression of *Axin* are concentrated to the presumptive dorsal endomesoderm and dorsal blastopore lip at the onset of gastrulation. (J) The dorsal part of blastopore of *Branchiostoma laceoltum* has a typical morphology and is different from the ventral part of blastopore at early gastrula stage. Dashed lines demarcate presumptive dorsal endomesoderm. (J') β-catenin is located in the dorsal part of blastopore at early gastrula stage and co-expressed with Goosecoid protein (J''). (K–K') The signal of β-catenin, which is labeled with specific anti-amphioxus β-catenin antibody, is located in the dorsal endomesoderm. (L) Axin is expressed at a higher level in the dorsal endomesoderm at early gastrula stage. (M–M') Enrichment of nuclear β-catenin and (N) *Axin* mRNA in the dorsal endomesoderm of the embryos at early cap-shaped mid-gastrula stage. (H–H'), (I), (M–N) Embryos are shown in the lateral view. (J–L) Blastopore view of the embryos. An, animal; Veg, vegetal; Dbl, dorsal blastopore lip; Bp, blastopore. Scale bar is 100 μM.

The online version of this article includes the following figure supplement(s) for figure 1:

**Figure supplement 1.** Immunodetection of nuclear β-catenin at the cleavage stage, late gastrula and late neurula stages Immunostaining of amphioxus embryos with antibodies against amphioxus.

*and A'-Ai'*). At this stage, the expression domains of Goosecoid and nuclear β-catenin markedly overlapped and located asymmetrically in the vegetal half of the embryo (*Figure 2A-Aiv'*). The location of both signals remained asymmetrical and overlapped during the blastula and the onset of gastrulation stages (*Figure 2B-Civ'*). At the early gastrula stage, both nuclear β-catenin and Goosecoid were present in the region of dorsal endomesoderm and the dorsal blastopore lip (*Figure 2D-Div'*). Additionally, weaker expression of Goosecoid protein was observed in the ventral endomesoderm (*Figure 2D–Di and D'Di''*), which is consistent with the expression of its mRNA at this stage (*Yu et al., 2007*). In contrast to the expression of nuclear β-catenin, the signal of which was mainly concentrated in the dorsal endomesoderm (*Figure 2Dii–Diii and Dii'-Diii'*), the expression of Goosecoid expanded to the dorsal ectoderm (*Figure 2D–Di and D'-Di'*). Similarly, at the cap-shaped gastrula stage, the strongest signal of nuclear β-catenin persisted in the dorsal endomesoderm (*Figure 2Eii–Eiii and Eii'-Eiii'*), while weaker expression of the Goosecoid protein was additionally present in the dorsal ectoderm (*Figure 2E–Ei and E'-Ei'*). Next, we performed immunostaining in the embryos of *Branchiostoma floridae* by using the same antibodies to detect β-catenin and

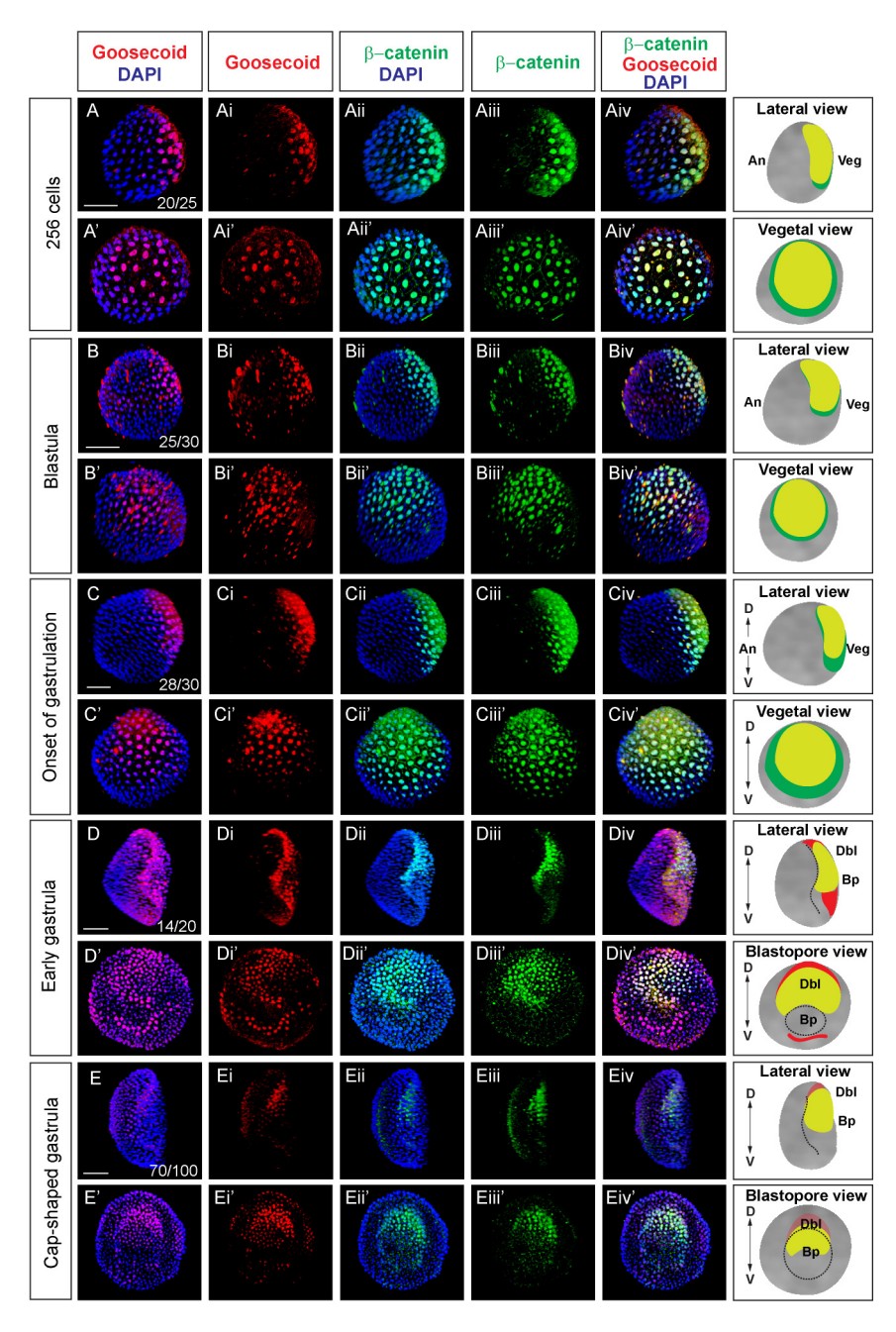

**Figure 2.** Nuclear β-catenin is colocalized with dorsal-specific protein Goosecoid during early development in amphioxus (A-Eiv') Double immunostaining of Gooosecoid and β-catenin in *Branchiostoma lanceolatum*. Lateral view of the embryos in (**A-Aiv**), (**B-Biv**), (**C-Civ**), (**D-Div**), (**E-Eeiv**). The embryos are shown from the vegetal pole view in (**A'-Aiv'**), (**B'-Biv'**), (**C'-Civ'**). Blastopore view of the embryos in (**D'-Div'**), (**E'-Eiv'**). Panels in the column on the right side represent diagrams of the embryos for a given developmental stage summarizing the expression of Goosecoid and β-catenin. D, dorsal; V, ventral; An, animal; Veg, vegetal; Dbl, dorsal blastopore lip; Bp, blastopore. Scale bar is 50 μM.

The online version of this article includes the following figure supplement(s) for figure 2:

**Figure supplement 1.** Nuclear β-catenin is colocalized with dorsal-specific protein Goosecoid and P-Smad2 during early development of *Branchiostoma floridae*.

Goosecoid. As in *Branchiostoma lanceolatum,* nuclear β-catenin and organizer-specific protein Goosecoid were asymmetrically distributed and their expression domains overlapped at the onset of gastrulation and early gastrula stages (*Figure 2—figure supplement 1A*).

Functional studies investigating the dorsal organizer formation in amphioxus presumed that Nodal signaling might be active in the dorsal domain of early amphioxus gastrula (*Le Petillon et al., 2017*; *Onai et al., 2010*). However, since no direct evidence exist to date to support this assumption, we performed immunostaining with anti-P-Smad2 antibodies. The specificity of anti-P-Smad2 antibodies was checked by immunostaining of the embryos at the mid-neurula stage, which were treated with inhibitor of P-Smad2-mediated signaling SB505124 (*Supplementary file 1*). We found that P-Smad2 is asymmetrically distributed and more P-Smad2-positive cells are present in the mesoderm on the left side (*Figure 3Aa-ai*), resembling the expression pattern of *Nodal*, *Lefty* and *Pitx* genes (*Soukup et al., 2015*). In addition, the treatment of the embryo with SB505124 conspicuously reduced the expression of P-Smad2 (*Figure 3Aaii-aiii*). Double immunostaining for Goosecoid protein and P-Smad2 revealed higher expression of P-Smad2 at the blastula and the onset of gastrulation stages in the domain where Goosecoid protein is expressed (*Figure 3Ba-biv'*). Further, we investigated the mutual spatial distribution of nuclear β-catenin and nuclear P-Smad2 proteins. Double immunostaining for β-catenin and P-Smad2 demonstrated that the both proteins are present in the embryo at the 64-cell stage (*Figure 3Ca-aiv'*). The intensity of nuclear β-catenin signal was substantially higher in one region of the vegetal half of the embryo (*Figure 3Ca-ai and Ca'-ai'*). In contrast, P-Smad2 signal was present in the entire nucleus, with slightly less intensity in the vegetal region and higher intensity in the animal half of the embryo (*Figure 3Caii-aiii and Caii'-aiii'*). At the early gastrula stage we detected the nuclear β-catenin and P-Smad2 in the same position in the embryo (*Figure 3Cb-biv'*). Additionally, nuclear β-catenin and P-Smad2 were expressed in the same domain in the embryos of *Branchiostoma floridae* at the onset of gastrulation (*Figure 2—figure supplement 1B*). Taken together, nuclear β-catenin overlapped with the expression of organizer-specific gene Goosecoid and mediator of Nodal signaling P-Smad2 (*Figure 3D*). These data allowed us to hypothesize that, similarly to vertebrates, Wnt/β-catenin might play a role in the establishment of dorsoanterior/ventroposterior axis and dorsal organizer in the basal chordate amphioxus.

## Wnt/β-catenin is required for the initial dorsal cell fate establishment in early amphioxus embryo

To find out whether Wnt/β-catenin signaling is required for the establishment of the dorsal organizer and the dorsoanterior/ventroposterior axis in amphioxus embryo, we first applied loss-of-function approaches by using pharmacological treatments with Porcupine inhibitor C59, which inhibits biosynthesis of Wnt ligands (*Supplementary file 1*). To test the efficiency of the treatments, we assayed Wnt/β-catenin signaling target gene *Axin* by in-situ hybridization for every individual treatment (*Figure 4A–Ai'*). The treatment of the embryos with C59 at one-cell stage resulted in strong suppression of dorsally specific genes *Chordin* and *Goosecoid* (*Figure 4B–Bi' and C-Ci'*), as well as the genes encoding ligands of the Nodal signaling pathway *Nodal* and *Vg1* (*Figure 4D–Di' and E-Ei'*). In addition, early neural genes *Elav* and *Neurogenin* were downregulated (*Figure 4F–Fi' and G-Gi'*). In contrast, the expression of ventrolateral specific gene *Vent1*, which is the direct target of Bmp signaling and functions as repressor of *Goosecoid* and *Chordin* genes, expanded dorsally upon C59 treatment (*Figure 4H–Hi'*). To complement pharmacological treatment we microinjected mRNA of dominant negative Tcf (dnTcf), which attenuates Wnt/β-catenin signaling by binding to cis-regulatory sequences of the target genes, but is unable to interact with β-catenin (*Figure 5A*). The injected embryos did not elongate properly, and dorsal structures such as notochord or neural tube were not recognizable at the mid-neurula stage (*Figure 5B and C*). The marker of neural tube and notochord *Chordin* was profoundly downregulated (*Figure 5Ba-a''*). Additionally, notochord marker Brachyury (*Figure 5Ca-a'''*) and neural tube marker Ac-Tub (*Figure 5Cb-b'''*) were not detected. Similarly, the expression of Elav disappeared from the neural tube and was only partially preserved in individual sensory cells in the ventral ectoderm (*Figure 5Cc-c'''*). These data suggest that the embryos injected with dnTcf lacked proper notochord and neural tube. However, we observed conspicuous expression of endodermal markers *Hex* (*Figure 5Bb-b''*) and FoxA (*Figure 5Cd-d'''*), which label the notochord and a considerable part of ventral endoderm. Taking into consideration that both markers of the notochord *Chordin* and Brachyury were strongly downregulated in the embryos injected with dnTcf, the FoxA-positive cells must only represent the endodermal expression of FoxA. Together, these

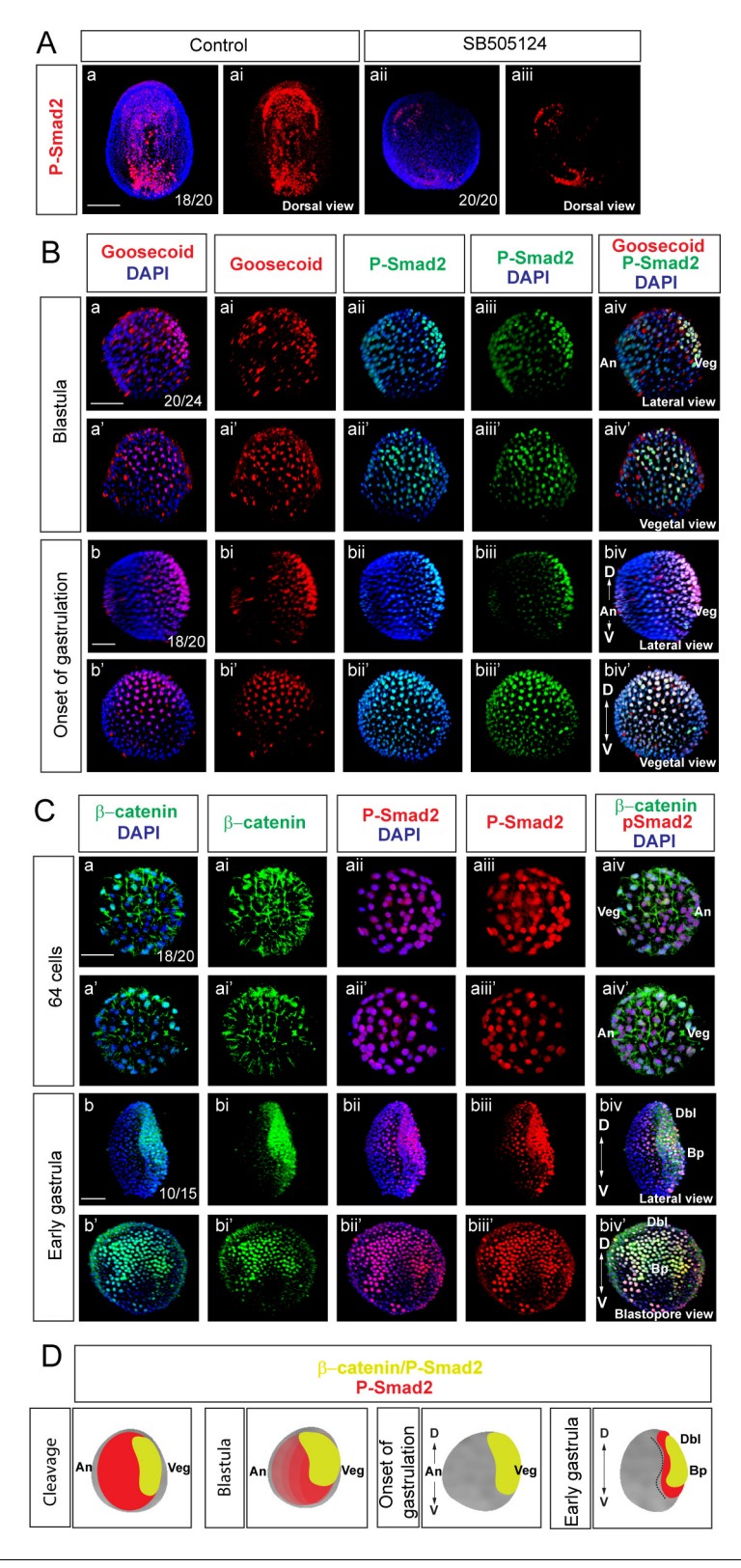

**Figure 3.** Wnt/β-catenin and Nodal signaling activities overlap during early amphioxus development. (A) Commercial anti-phosphoSmad2 antibodies specifically label phospho-Smad2 (P-Smad2) in amphioxus. (B) Double immunostaining of Goosecoid and P-Smad2. (C) Double immunostaining of P-Smad2 and β-catenin during early development of *Branchiostoma lanceolatum.* (D) Schematic diagrams of the embryos summarizing the expression

*Figure 3 continued on next page*

*Figure 3 continued*

of P-Smad2 and β-catenin. (**Aa-Aai**) Dorsal view of early amphioxus neurula demonstrates stronger signal of P-Smad2 in the left endoderm and mesoderm. (**Aa-Aaiii**) Treatment with P-Smad2-mediated signaling pathway inhibitor SB505124 downregulates the expression of P-Smad2. (**B**) Goosecoid and P-Smad2 are co-expressed at the blastula (**Ba-Baiv'**) and onset of gastrulation (**Bb-Bbiv'**) stages. Red dots that do not correspond to nuclei represent nonspecific random sticking of antibodies on the surface of the embryos. (**C**) The expression of nuclear β-catenin and P-Smad2 at 64-cell stage (**Ca-Caiv'**) and early gastrula stage (**Cb-Cbiv'**). (**A**) Dorsal view of the embryo. (**Ba-Baiv**), (**Bb-Bbiv**), (**Cb-Cbiv**) and (**Da-Daiv**) Lateral view of the embryos. The embryos are shown from the view of the vegetal pole in (**Ba'-Baiv'**), (**Bb'-Bbiv'**) and (**Da'-Daiv'**). Blastopore view in (**Cb'-Cbiv'**). (**Ca-Caiv**) Two sides of the view with animal pole to the right in (**Ca-Caiv**) and to the left in (**Ca'-Caiv'**). D, dorsal; V, ventral; An, animal; Veg, vegetal; Dbl, dorsal blastopore lip; Bp, blastopore. Scale bar is 50 µM.

results demonstrate that Wnt/β-catenin signaling is required for the expression of dorsal-specific genes during the initial stages of amphioxus embryonic development.

## Ectopic activation of Wnt/β-catenin signaling after fertilization but not at the blastula stage promote expression of dorsal-specific genes

To support the hypothesis that Wnt/β-catenin regulates dorsal cell fate in early amphioxus development, we administered Wnt/β-catenin signaling activator CHIR99021, which inhibits phosphorylation and degradation of β-catenin (*Supplementary file 1*). We treated the embryos at two distinct time points with a low (5 µM) and high (20 µM) concentration of CHIR99021 (*Figure 6A and B*) and tested the expression of dorsal-specific genes *Chordin, Goosecoid* or *Nodal*. Administration of low and high concentrations of the drug after fertilization caused upregulation of Wnt/β-catenin signaling target gene *Axin* (*Figure 6Aa-b', Ah-I', Ba-b', Bi-j'*). The low dosage of CHIR99021 provoked ectopic expression of *Chordin* throughout the blastopore ring (*Figure 6Ac-e'*) and expansion of the *Goosecoid* expression domain (*Figure 6Af-g'*). Administration of a high concentration of CHIR99021 after fertilization caused morphological defects in the embryos at the mid-gastrula stage (*Figure 6Ah-n'*). The embryos did not invaginate properly, suggesting that strong overactivation of Wnt/β-catenin signaling might disrupt the process of gastrulation. In these embryos, the expression of *Chordin* and *Nodal* spread out over the whole embryo (*Figure 6Aj-k' and Al-n'*). Although the embryos treated with high concentration of CHIR99021 at one-cell stage did not invaginate properly the blastopore ring was formed and partial invagination of vegetal plate took place. We did not observe the formation of exogastrula and strong vegetalization phenotype as described previously for sea urchin (*Wikramanayake et al., 1998*).

When we treated the embryos with a low concentration of CHIR99021 at the blastula stage, we did not observe ventral or lateral expansion of the *Goosecoid* and *Chordin* expression at the early neurula stage (*Figure 6Bc-d' and Bf-g'*). However, the expression domains of Chordin and Goosecoid were expanded anteriorly and posteriorly, suggesting that Wnt/β-catenin signaling plays a role in the establishment of the anterior/posterior axis after blastula stage (*Figure 6Bc-d' and Be'*). These data are consistent with previous observations (*Holland et al., 2005*; *Onai et al., 2009*). Administration of a high concentration of CHIR99021 at the blastula stage led to downregulation of the *Chordin* and *Goosecoid* expression (*Figure 6Bk-l' and Bn-o'*), indicating that Wnt/β-catenin signaling does not promote the dorsal cell fate after the cleavage stage. Taken together, loss-of-function and gain-of-function experiments demonstrated that Wnt/β-catenin is essential for the expression of dorsal-specific genes and might play an important role in the initiation of the dorsal cell fate in early amphioxus embryo.

## Ectopic activation of Wnt/β-catenin signaling induces ectopic axis in cephalochordate amphioxus

Next, we were interested in whether ectopic activation of Wnt/β-catenin signaling induces phenotypic changes of the amphioxus embryos at later stages of development, when notochord and neural tube are forming. Therefore, we performed mRNA microinjection of Wnt ligands (*Wnt8* and *Wnt11*) into the amphioxus egg or treatment of embryos with CHIR99021 and scored the phenotype and protein or mRNA expression of notochord marker *Brachyury* and neural markers *Elav* or Ac-Tub (acetylated tubulin) at the neurula or larval stage. We assumed that by microinjection of Wnt ligand

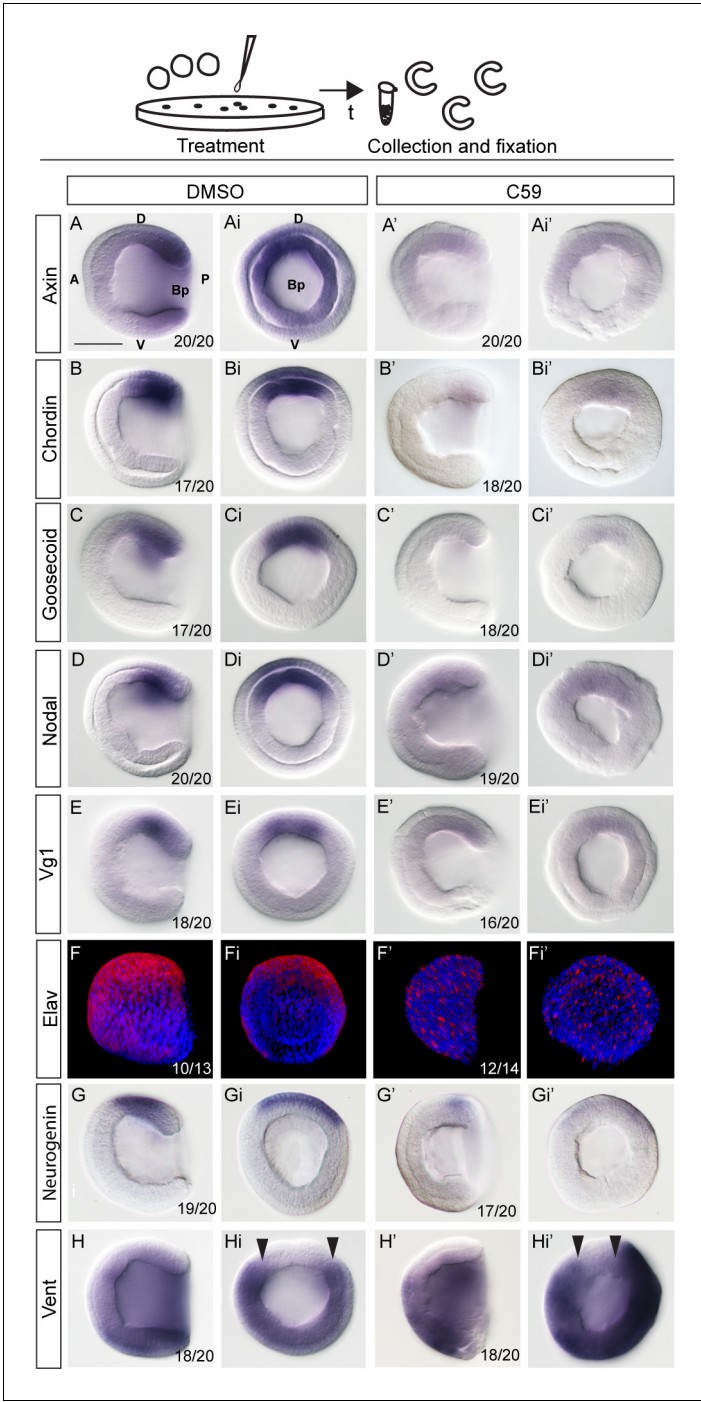

**Figure 4.** Treatment of amphioxus embryos with the inhibitors of Wnt/β-catenin signaling at one-cell stage results in downregulation of the dorsal-specific genes at the gastrula stage. The expression of Wnt/β-catenin target *Axin*, dorsal-specific genes *Chordin* and *Goosecoid*, dorsal-specific genes encoding signaling molecules *Nodal* and *Vg1*, dorsal-specific neural markers *Neurogenin* and *Elav*, ventral-specific gene *Vent1* in the control DMSO-treated embryos and embryos treated continuously with Wnt/β-catenin signaling inhibitor C59 after fertilization at one-cell stage. The embryos are at mid-gastrula stage. Lateral view in (**A–H and A'–H'**) and blastopore view in (**Ai-Hi and Ai'-Hi'**). Arrowheads in Hi and Hi' mark the extent of dorsal Vent expression. D, dorsal; V, ventral; A, anterior; P, posterior; Bp, blastopore. Scale bar is 50 μM.

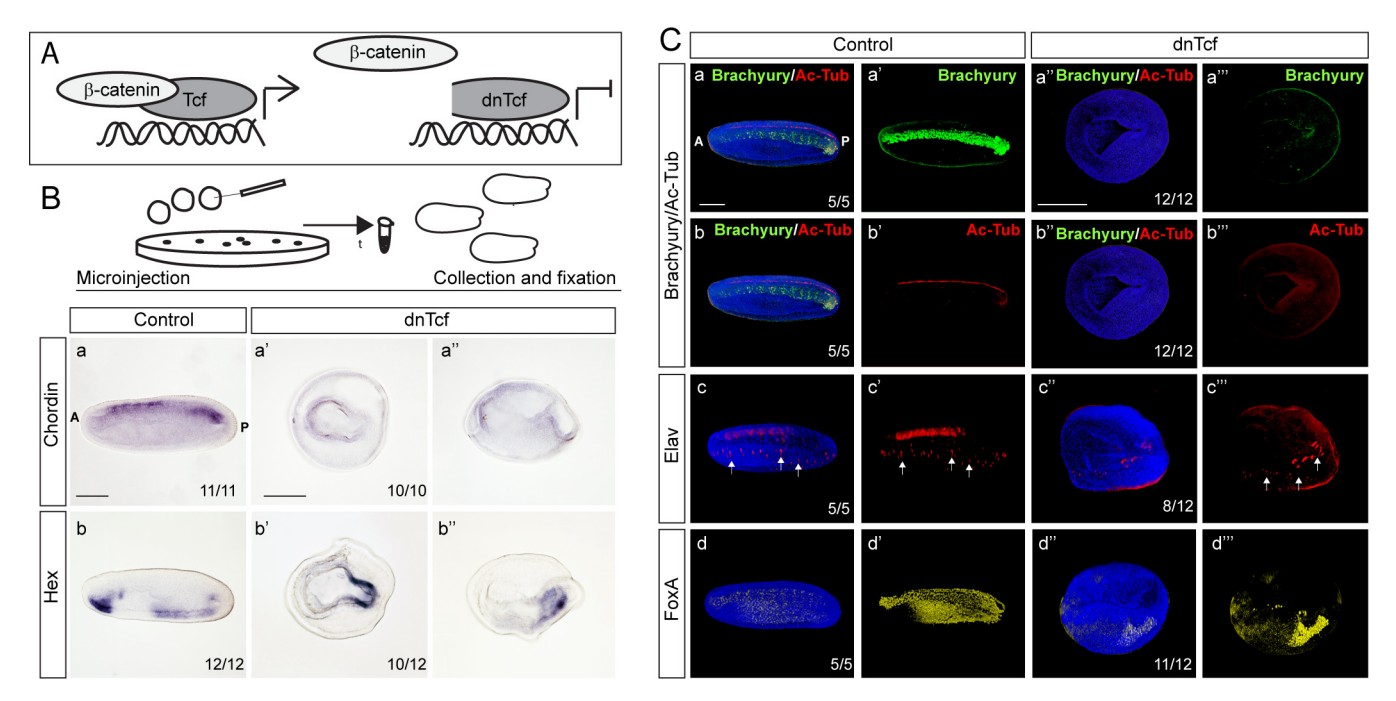

**Figure 5.** Inhibition of Wnt/ß-catenin signaling by injection of dominant-negative Tcf (dnTcf) mRNA leads to impairment of the axis development in amphioxus neurula. (**A**) The scheme illustrates the mechanism of Wnt/ß-catenin signaling inhibition by dnTcf. (**B**) mRNA expression of Chordin and Hex genes in the control embryos and embryos injected with dnTcf. (**C**) Protein expression of Brachyury, acetylated tubulin (Ac-Tub), Elav and FoxA in the control embryos and embryos injected with dnTcf. In (**Ca**), (**Cb**) and (**Ca''**), (**Cb''**) the same embryos with double expression of Brachyury and Ac-Tub are shown. White arrows demarcate isolated sensory cells. In (**Ba**), (**Bb**), (**Ca-a'**), (**Cb-b'**), (**Cc-c'**) and (**Cd-d'**) the embryos are shown in lateral view. In (**Ba'**), (**Bb'**), (**Ca''-a'''**), (**Cb''-b'''**), (**Cc''-c'''**) and (**Cd''-d'''**) the embryos are shown in blastopore view. Anterior to the left. A, anterior; P, posterior. Scale bar is 100 μm.

mRNA we would be able not only to overactivate Wnt/β-catenin signaling, but also randomly disrupt or change the activity signaling gradient in the early embryo. Although *Wnt8* is not expressed in the organizer of vertebrates and amphioxus, injected Wnt8 mRNA promotes ectopic axis formation in the vertebrates (*Smith and Harland, 1991*; *Sokol et al., 1991*). When we injected *Wnt8* mRNA at a low dose, most embryos had a mild phenotype (*Figure 7A b-d' and Af-g'*, compare to *Figure 7Aa-a' and Ae-e'* and *Figure 7—figure supplement 1*) and displayed weak changes in the expression of Brachyury (*Figure 7A b-d'*, compare to *Figure 7Aa'*). In addition, the low concentration of *Wnt8* mRNA did not result in conspicuous changes of Elav expression in the neural plate (*Figure 7A f-g'* compare to *Figure 7Ae-e'*). Higher concentration caused a severe phenotype in the embryos at the neurula stage (*Figure 7Ai–k' and Am-o'*, compare to *Figure 7Aa-a' and Ae-e'* and *Figure 7—figure supplement 1*) and the expression of Brachyury and Elav was disrupted (*Figure 7Ai–k' and Am-o'*). In certain embryos, the expression pattern of Brachyury and Elav was still recognizable in the presumed dorsal domain (*Figure 7Ai–i' and Am-m'*). However, in the majority of the embryos, the correct expression along the dorsal axis was lost and two randomly located domains of Brachyury and Elav expression were formed (*Figure 7A j-k' and An-o'*).

Further, we tested whether we are able to induce a similar phenotype in amphioxus embryos by treatment with CHIR99021. The embryos that were treated with CHIR99021 at the concentration of 10 μM at the one-cell stage gastrulated properly and survived until the larval stage, but the larvae had severe phenotypic changes without clearly defined dorsoanterior/ventroposterior axis (*Figure 7Bb-d and Bf-h*, compare to *Figure 7Ba and Be*). The longitudinal stripes of *Brachyury* expression were spread throughout the embryos (*Figure 7Bb-c* compare to *Figure 7Ba*). In some embryos, the expression was stronger in one region (*Figure 7Bb*) or spread almost over the whole embryo (*Figure 7Bd*). We did not observe extensive expansion of *Elav* expression (*Figure 7Bf*), and two expression domains appeared only in certain embryos (*Figure 7Bg-h*). The treatment of the

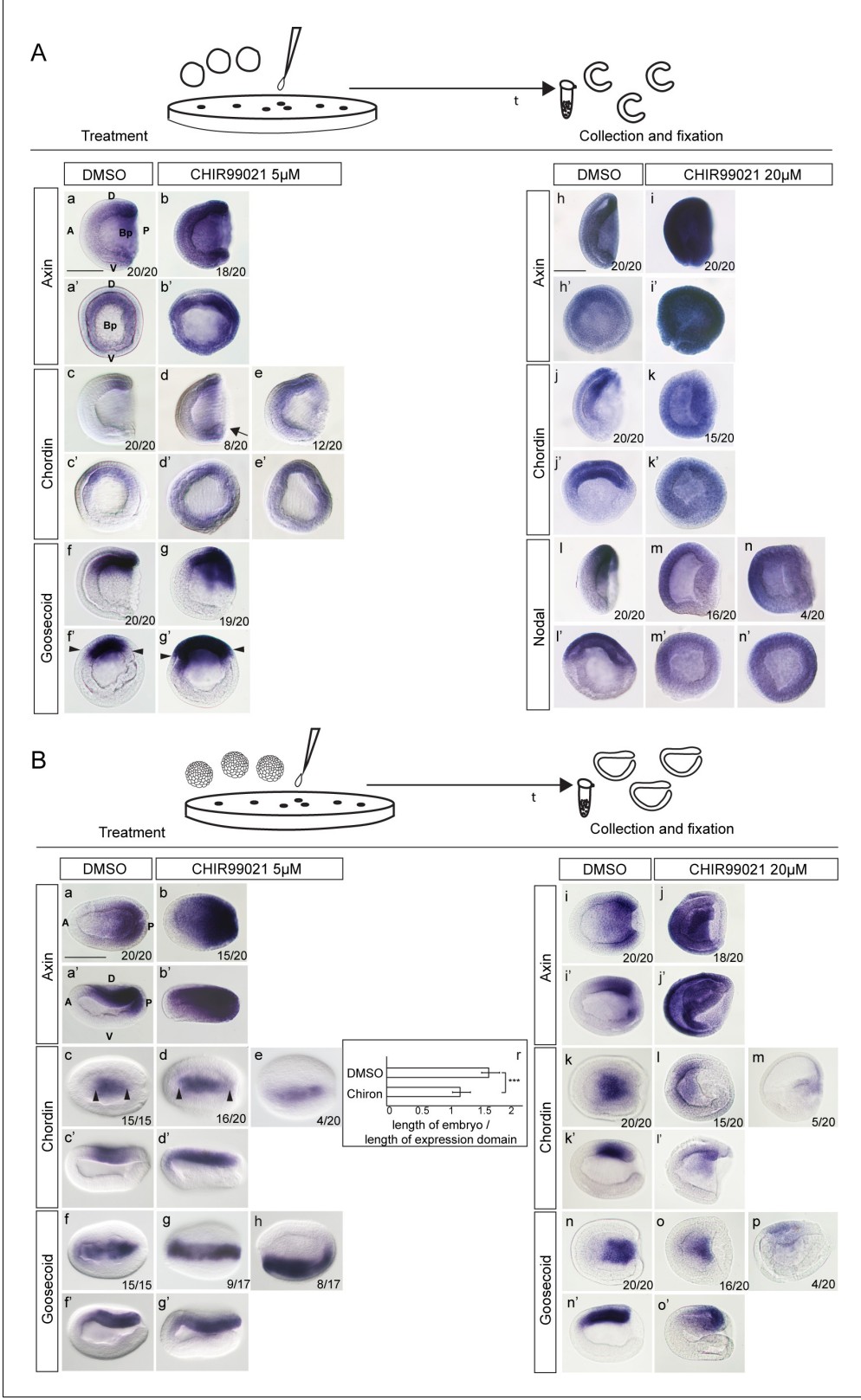

**Figure 6.** Ectopic activation of Wnt/β-catenin signaling at one-cell stage but not at the blastula stage results in ectopic expression of dorsal-specific genes (**Aa-g'**). The expression of *Axin*, *Chordin* and *Goosecoid* in the control embryos and embryos treated with 5 μM of CHIR99021 at one-cell stage. Arrow in Ad marks ventral expansion of Chordin expression. Arrowheads in Af' and Ag' mark the extent of Goosecoid expression. (**Ah-n'**) The expression

*Figure 6 continued on next page*

*Figure 6 continued*

of *Axin*, *Chordin* and *Nodal* in the control embryos and embryos treated with 20 µM of CHIR99021 at one-cell stage. (**Ba-g'**) The expression of *Axin*, *Chordin* and *Goosecoid* in the control embryos and embryos treated with 5 µM of CHIR99021 at the blastula stage. Arrowheads in Bc and Bd mark the extent of Chordin expression. (**Bi-o'**) The expression of *Axin*, *Chordin* and *Goosecoid* in the control embryos and embryos treated with 20 µM of CHIR99021 at the blastula stage. (**Br**) Chart illustrating significant posterior and anterior extension of the *Chordin* expression domain in the treated embryos (p<0.01, paired t-test). (**Aa, Ab, Ac, Ad, Af, Ag, Ah, Ai, Aj, Ak, Al, Am, An, Ba', Bb', Bc', Bd', Bf', Bg', Bi', Bj', Bk', Bl', Bn', Bo'**) The embryos are shown in lateral view. (**Aa', Ab', Ac', Ad', Ae', Af', Ag', Ah', Ai', Aj', Ak', Al', Am', An'**) The embryos are shown in blastopore view. (**Ba, Bb, Bc, bd, Be, Bf, Bg, Bh, Bi, Bj, Bk, Bl, Bm, Bn, Bo, Bp**) The embryos are shown in dorsal view. D, dorsal; V, ventral; A, anterior; P, posterior; Bp, blastopore. Scale bar is 100 µM.

embryos with CHIR99021 at the blastula stage led to the development of larvae with a mild phenotype exhibiting shortened head and undeveloped rostrum (*Figure 7Cb and Cd*, compare to *Figure 7Ca and Cc*), consistent with previous findings (*Holland et al., 2005*; *Onai et al., 2009*) and evolutionary conservation (*Petersen and Reddien, 2009*) of Wnt/β-catenin signaling in anterior/posterior patterning. In the treated larvae, *Brachyury* and *Elav* genes were dorsally expressed along the embryo in a similar manner as in the control larvae (*Figure 7Cb and Cd*, compare to *Figure 7Ca and Cc*).

*Wnt11* was detected by RT-PCR at the one-cell stage embryo (*Qian et al., 2013*) and expressed asymmetrically in the vegetal half of the amphioxus blastula (*Somjai et al., 2018*) and dorsally at the onset of gastrulation and mid-gastrula stage (*Schubert et al., 2000a*). Microinjection of *Wnt11* mRNA at a low dose caused conspicuous phenotypical changes in approximately one-third of the injected embryos at the late neurula stage (*Figure 8Ab-d''*, compare to *Figure 8Aa-a''*). In certain embryos, both Brachyury and Ac-Tub were ectopically expressed, suggesting formation of a secondary axis (*Figure 8Ab-b''*, compare to *Figure 8Aa-a''*). In other embryos, although Brachyury expression was changed (*Figure 8Ac'*), two regions of expression arose just in the case of Ac-Tub (*Figure 8Ac''*). However, most embryos did not exhibit changes in the phenotype and ectopic expression of the notochord or neural tube markers (*Figure 8Ad-d''*). Only after microinjection of a high dose of *Wnt11* mRNA did we observe strong changes of the phenotype in all embryos (*Figure 8Ba-e''*). Merely half of the embryos demonstrated formation of a secondary axis (*Figure 8Bc-d''*). In a few embryos, we observed a small region of ectopic Brachyury and Act-tub expression in the ventral part of the embryo (*Figure 8Be-e''*). Taken together, these data suggest that strong ectopic activation of Wnt/β-catenin signaling by injection of Wnt ligand mRNA leads to the formation of two regions of presumptive dorsal axis in the amphioxus embryos. Combined, these results indicate that ectopic activation of Wnt/β-catenin signaling at the one-cell stage causes severe impairment of amphioxus dorsoanterior/ventroposterior axis development. In contrast, overactivation of Wnt/β-catenin signaling at the mid-blastula stage does not influence DV axial patterning during the amphioxus development.

## Nodal and Wnt/β-catenin signaling cooperate to regulate the establishment of the dorsal cell fate and dorsoanterior/ventroposterior axis in the amphioxus

To examine the possibility of cooperative functioning of Nodal and Wnt/β-catenin signaling in the dorsal cell fate specification of amphioxus embryos, we treated the embryos after fertilization with low concentrations of inhibitor of Wnt/β-catenin signaling C59 (1 µM) and Nodal receptor inhibitor SB505124 (0.5 µM) separately and simultaneously. We left the embryos to develop until the cap-shaped gastrula to analyze the expression of dorsal-specific genes *Goosecoid* and *Nodal*, which marks the dorsal organizer at this stage and at the same time encodes the ligand of the Nodal signaling pathway (*Figure 9A*), or mid-neurula stage to score the expression of notochord marker *Brachyury* and neural marker *Elav* (*Figure 9B*). Separate administration of low concentrations of C59 and SB505124 caused only slight downregulation of *Nodal* (*Figure 9Aai-ai' and Aaii-aiii*, compare to *Figure 9Aa-ai*) and *Goosecoid* (*Figure 9Abi-bi' and A bii-bii'*, compare to *Figure 9A b-b'*) and did not cause changes in the morphology of the embryos at the mid-neurula stage (*Figure 9B*). The expression of *Brachyury* remained unchanged after C59 treatment (*Figure 9Bai'-ai'B*, compare to

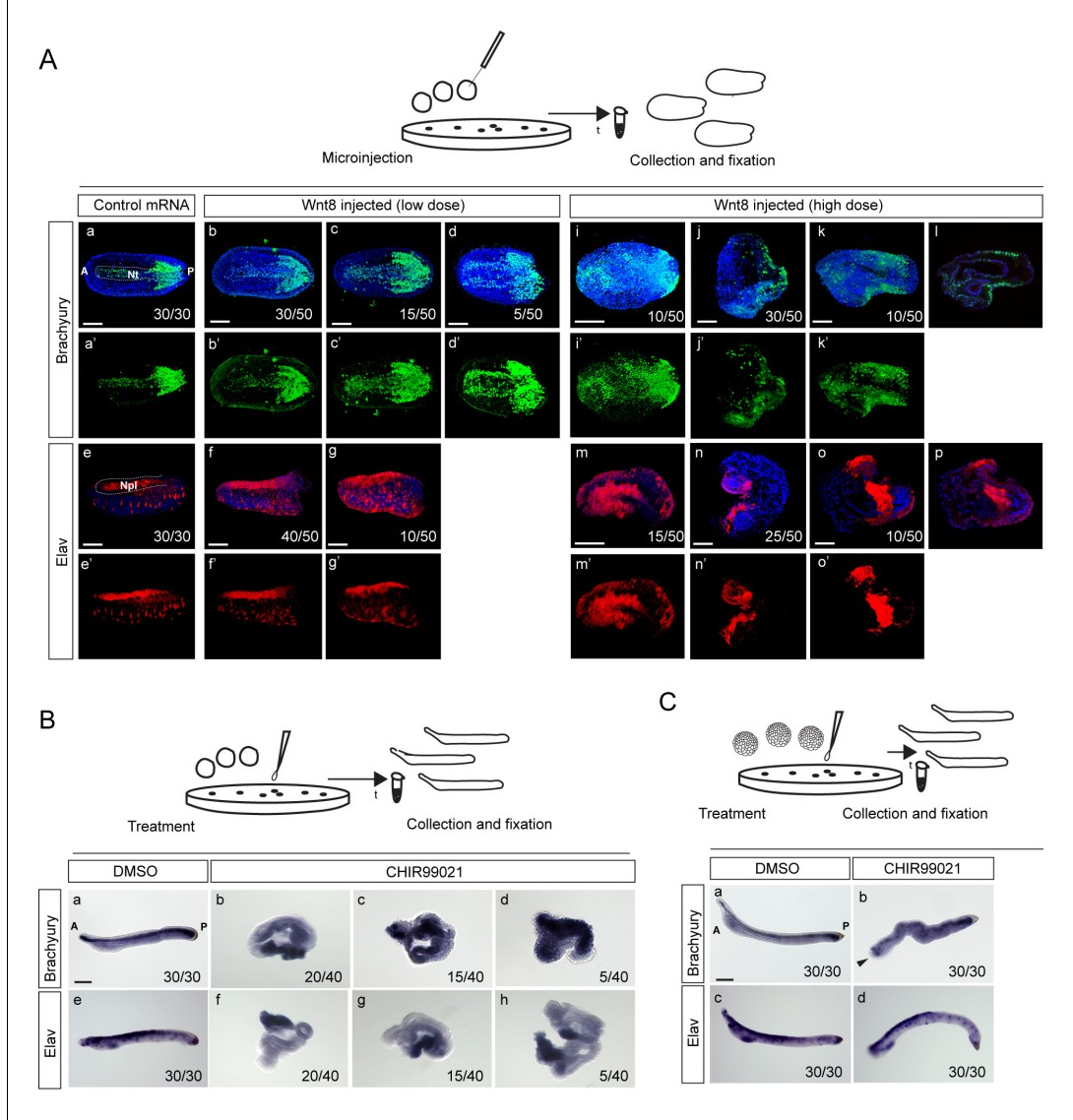

**Figure 7.** Ectopic activation of Wnt/β-catenin signaling pathway by microinjection of Wnt8 ligand mRNA or treatment with CHIR99021 induces ectopic expression of dorsal axis-specific markers in the amphioxus embryos. (**A**) Expression of Brachyury and Elav proteins at the mid-neurula of amphioxus embryos injected with *Wnt8* mRNA at a low dose (3 ng/µl) or high dose (7 ng/µl). (**B–C**) Expression of *Brachyury* and *Elav* mRNA at the larval stage of amphioxus embryos treated with 10 µM of CHIR99021 at one-cell stage (**B**) or at blastula stage (**C**). Arrowhead in Cb marks the truncation of the head. The embryos are shown in dorsal view in (**Aa-Ad′**), (**Ai-i′**) and (**Am-m′**). Lateral view in (**Ae-g′**), (**Ba**), (**Be**) and (**Ca-d**). (**Al**) and (**Ap**) represent the individual z-stacks of (**Ak**) and (**Ao**), respectively. A, anterior; P, posterior; Nt, notochord; Npl, neural plate. Scale bar is 100 µM.

The online version of this article includes the following figure supplement(s) for figure 7:

**Figure supplement 1.** Wide-field image of amphioxus embryos injected with tdTomato mRNA or tdTomato with Wnt8 mRNA.

*Figure 9B a-a′*) and was only slightly downregulated after SB505124 treatment (*Figure 9B aii-aii′*, compare to *Figure 9B a-a′*). The expression of neural marker *Elav* was moderately upregulated at the late neurula stage in the embryos treated with a low concentration of C59 (*Figure 8Bbi-bi′* compare to *Figure 8B b-b′*), which might be related to the later function of Wnt/β-catenin signaling in promoting the ventral and posterior cell fate during gastrulation and after blastula stage of development (*Onai et al., 2009*). The treatment with a low concentration of Nodal signaling inhibitor SB505124 led to slight shortening of the *Elav* expression domain (*Figure 9Bbii-bii′*, compare to *Figure 9B b-b′*). When the drugs were applied simultaneously, the expression of *Nodal* and *Goosecoid* was suppressed to a greater extent than when the drugs were applied separately

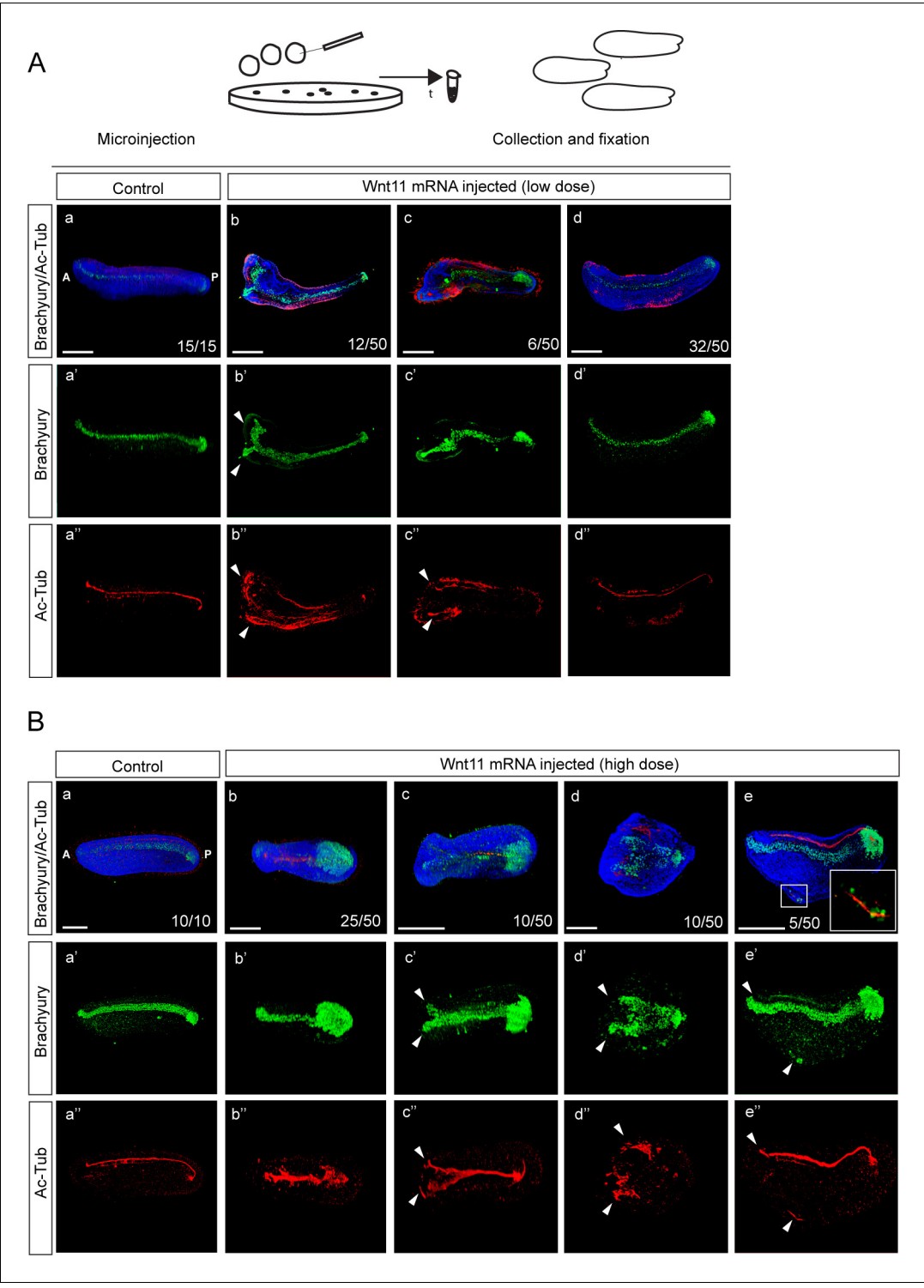

**Figure 8.** Ectopic activation of Wnt/β-catenin signaling pathway by microinjection of *Wnt11* ligand mRNA induces formation of a double axis in the amphioxus embryos. Double immunostaining detecting the protein expression of Brachyury and Ac-Tub in the embryos injected with a low (**A**) or high (**B**) dose of *Wnt11* mRNA (50 ng/μl or 250 ng/μl, respectively). The insert in (Be) demonstrates the enlarged region from the ventral part of the embryo. Embryos are at early larval (Aa-d'') or the mid-neurula stage (Ba-e''). The embryos are shown in lateral view in (Aa-a''), (Ad-d''), (Ba-a'') and (Be-e''). The embryos are shown in dorsal view in (Ab-c'') and (Bb-d''). Arrowheads mark the twinning of notochord and neural tube following the injection of *Wnt11* mRNA. A, anterior; P, posterior. Scale bar is 100 μM.

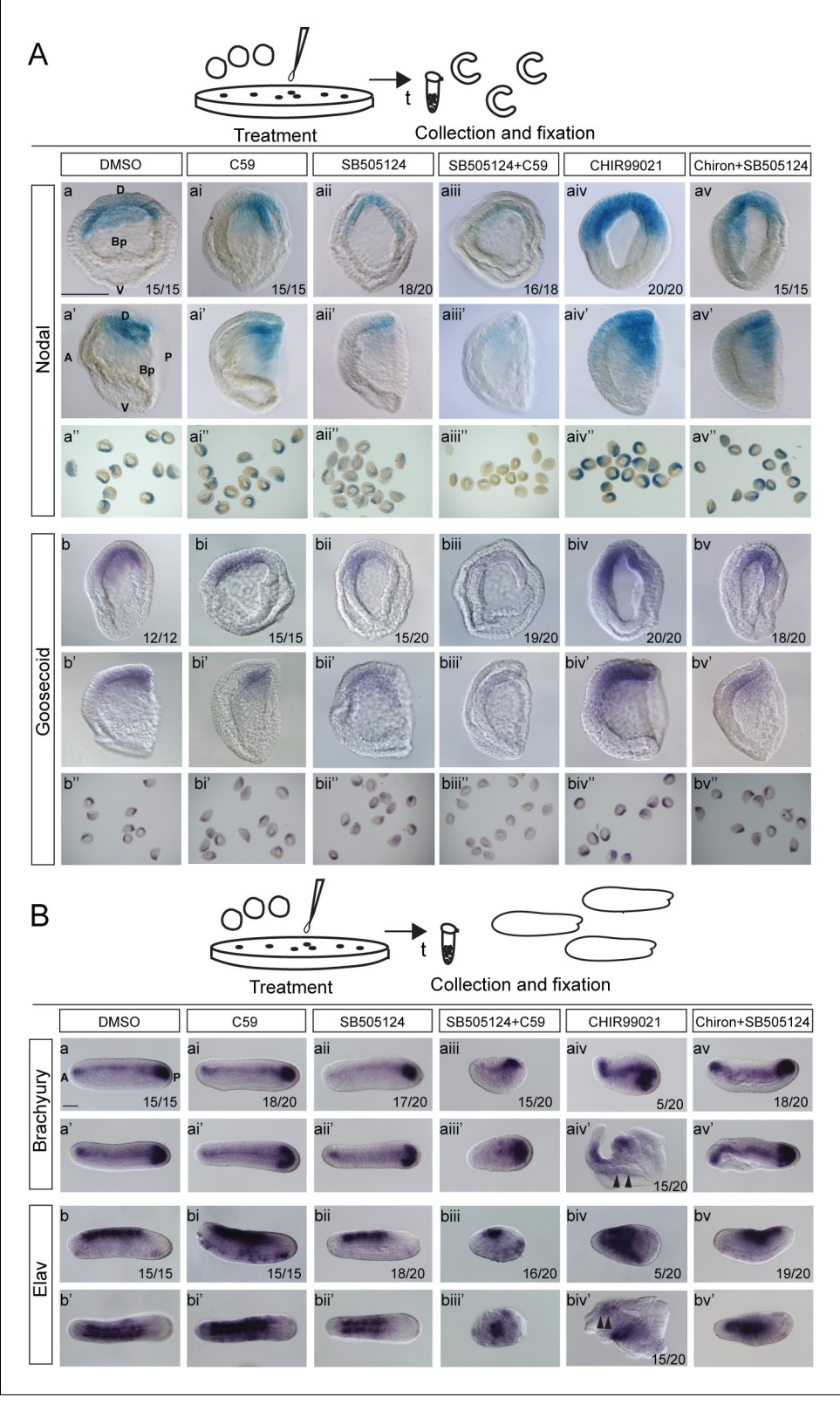

**Figure 9.** Wnt/β-catenin and Nodal signaling pathways co-act during early development to promote dorsal cell fate in the amphioxus embryo. (**A**) Expression of *Nodal* (**Aa-av''**) and *Goosecoid* (**Ab-bv''**) in the control embryos, which were treated with DMSO (**Aa-a'' and Ab-b''**), and the embryo treated with Wnt signaling inhibitor C59 (**Aai-ai'' and Abi-bi''**), Nodal signaling inhibitor SB505124 (**Aaii-aii'' and Abii-bii''**) separately or simultaneously (**Aaiii-**

*Figure 9 continued on next page*

*Figure 9 continued*

aiii'' and Abiii-biii''), Wnt/β-catenin signaling activator CHIR99021 (**Aaiv-aiv''** and **Abiv-biv''**) separately or simultaneously with SB505124 (**Aav-av''** and **Abv-bv''**) at one-cell stage. The embryos are at the mid-gastrula stage. (**B**) Expression of *Brachyury* (**Ba-av'**) and *Elav* (**Bb-bv'**) in the control embryos, which were treated with DMSO (**Ba-a'** and **Bb-b'**), and the embryo treated with Wnt signaling inhibitor C59 (**Bai-ai'** and **Bbi-bi'**), Nodal signaling inhibitor SB505124 (**Baii-aii'** and **Bbii-bii'**) separately or simultaneously (**Baiii-aiii'** and **Bbiii-Biii'**), Wnt/β-catenin signaling activator CHIR99021 (**Baiv-aiv'** and **Bbiv-biv'**) separately or simultaneously with SB505124 (**Bav-av'** and **Bbv-bv'**) at one-cell stage. The embryos are at the late neurula stage. Arrowheads in Baiv' and Bbiv' mark weak ectopic expression of *Brachyury* and *Elav*, respectively. (**Aa-av, Ab-bv**) The embryos are shown from blastopore view. (**Aa'-av', Ab'-bv', Ba-av, Bb-bv**) The embryos are shown in lateral view. (**Ba'-av', Bb'-bv'**) The embryos are shown in dorsal view. D, dorsal; V, ventral; A, anterior; P, posterior; Bp, blastopore. Scale bar is 50 μM.

The online version of this article includes the following figure supplement(s) for figure 9:

**Figure supplement 1.** Assaying the responsiveness of cis-regulatory elements of amphioxus *Goosecoid*, *Nodal* and *Vg1* to Wnt/β-catenin.

---

(*Figure 9Aaiii-aiii''*, compare to *Figure 9Aai-ai'* and *Figure 9Aaii-aii'*; *Figure 9Abiii-biii'*, compare to *Figure 9Abi-bi''* and *Abii-bii'*). In addition, co-administration of C59 and SB505124 caused strong phenotypic changes at the mid-neurula stage (*Figure 9Baiii-aiii' and Bbiii-biii'*, compare to *Figure 9Ba-a' and Bb-b'*). In the treated embryos, *Brachyury* was not expressed in the dorsomedial position of the embryo and only a posterior patch of *Brachyury* expression, which marks posterior dorsolateral mesoderm, was preserved (*Figure 9Baiii-aiii'*, compare to *Figure 9Ba-a'*), indicating that the notochord was not developed properly over the dorsal axis. However, co-inhibition of Wnt/β-catenin and Nodal signaling did not cause complete loss of *Elav* expression (*Figure 9Bbiii-biii'*). Further, we examined the combined effect of overactivation of Wnt/β-catenin signaling and inhibition of Nodal signaling. Administration of low dose of CHIR99021 caused slight but noticeable upregulation of *Nodal* and *Goosecoid* genes at the cap-shaped gastrula stage (*Figure 9Aaiv-aiv''*, compare to *Figure 9Aa-a''* and *Figure 9Abiv-biv''*, compare to *Figure 9Ab-b''*) and conspicuous phenotypic changes in the embryos at the neurula stage (*Figure 9Baiv-aiv' and Bbiv-biv'*, compare to *Figure 9Ba and Bb*). The expression of *Brachyury* was either slightly upregulated (*Figure 9Baiv*), or the ectopic domain of weak expression originated in the majority of the embryos that did not preserve the properly formed DV axis (*Figure 9Baiv'*). *Elav* was expressed over the whole dorsal ectoderm in the embryos with the mild phenotype (*Figure 9Bbiv*), or its expression was spread over a considerable part of the embryo with one small strong and one weaker distinct domain (*Figure 9Bbiv'*). Remarkably, the simultaneous application of inhibitor of Nodal signaling SB505124 and activator of Wnt/β-catenin signaling CHIR99021 did not noticeably change the expression of *Nodal* and *Goosecoid* at the cap-shaped gastrula stage (*Figure 9Aav-av'' and Abv-bv''*, compare to *Figure 9Aa-a''* and *Figure 9Ab-b''*) and caused mild phenotypic changes in the embryos at the neurula stage (*Figure 9Bav-av' and Bbv-bv'*) as compared to the control embryos (*Figure 9Ba-a' and Bb-b'*) or the embryos treated with CHIR99021 only (*Figure 9Baiv-aiv' and Bbv-bv'*). Further, the expression of *Brachyury* and *Elav* was only partially affected in these embryos (*Figure 9Bav-av' and Bbv-bv'*). Combined, these results show that inhibition of Nodal signaling can rescue the strong phenotypic defects that have resulted from the ectopic activation of Wnt/β-catenin signaling. Together, these data demonstrate that Nodal and Wnt/β-catenin signaling cooperate early in development to initiate the dorsal cell fate and establish the correct dorsoanterior/ventroposterior axis in the amphioxus embryo. Moreover, their cooperation is important for promoting the Nodal signaling itself.

## Discussion

### Asymmetrical Wnt/β-catenin signaling specifies early embryonic polarity and plays a role in the early embryonic organizer formation in metazoans

In many metazoans, nuclear β-catenin is distributed asymmetrically during the first cleavage stages and involved in determination of primary polarity of the early embryo, which consequently has an influence on the germ layer specification and establishment of the body axis. In various invertebrates

such as nematodes, annelids, nemertean, mollusks, sea urchins, and ascidians β-catenin is stabilized and nuclearized in the vegetal blastomeres and plays an important role in the gastrulation and endoderm or endomesoderm formation (*Rocheleau et al., 1997*; *Logan et al., 1999*; *Henry et al., 2010*; *Henry et al., 2008*; *Schneider and Bowerman, 2007*; *Imai et al., 2000*). Remarkably, the asymmetrical distribution of Wnt/β-catenin signaling components and their involvement in the establishment of primary body axis was also reported in *Hydra* (*Hobmayer et al., 2000*) and sea anemone *Nematostella vectensis* (*Henry et al., 2010*; *Kraus et al., 2016*), the members of the bilaterian sister group Cnidaria. These findings suggest deep evolutionary conservation of the mechanism involving Wnt/β-catenin-dependent initiation of the embryonic polarity and axis establishment throughout the animal kingdom. In light of this finding, our study demonstrating that the Wnt/β-catenin pathway plays a key role in the organizer formation of basal chordate amphioxus is really important in arguing that this mechanism operates across the Metazoa (*Figure 10*).

In vertebrate zebrafish, β-catenin is first detectable in the nuclei of dorsal blastomeres at the 128-cell stage (*Dougan et al., 2003*). In *Xenopus*, nuclear β-catenin is observed as early as the 16-cell stage, and then is enriched at the 32-cell stage in dorsal vegetal blastomeres (*Larabell et al., 1997*), which form the Nieuwkoop center at the early blastula stage. Similarly, in cephalochordate amphioxus we first detected asymmetrical distribution of nuclear β-catenin at the 32-cell cleavage stage in the vegetal blastomeres. By the 64-cell stage, β-catenin is accumulated in one third of the vegetal

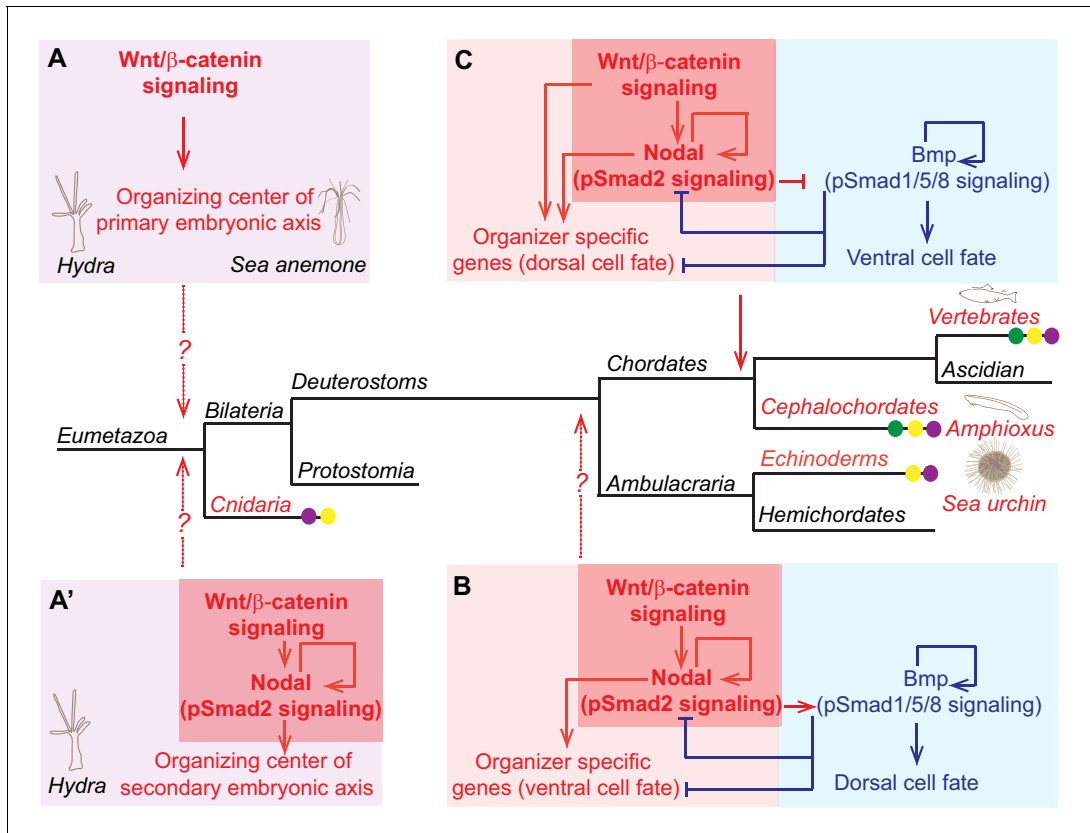

**Figure 10.** The role of Wnt/β-catenin signaling in the establishment of axial patterning in Metazoa. (A) Wnt/β-catenin signaling determines the organizing center of the primary body axis formation in cnidarian *Hydra* and *Nematostella*. (A') Wnt/β-catenin upstream of autoactivating Nodal signaling acts to induce a secondary axis formation in cnidarian *Hydra*. (B) Wnt/β-catenin, Nodal and Bmp signaling specify the secondary ventral/dorsal (also called oral/aboral axis) in sea urchin. Note that according to the hypothesis of dorso-ventral inversion occurred in chordate lineage (*Arendt and Nübler-Jung, 1994*) the establishment of ventral/dorsal axis in sea urchin corresponds to the establishment DV axis in chordates. (C) Wnt/β-catenin, Nodal and Bmp signaling specify the secondary DV axis in chordates. (A'), (B) and (C) Core-signaling cassette involving Wnt/β-catenin upstream of autoactivating Nodal signaling (highlighted by the red boxes) acts to induce a secondary axis formation in cnidarian *Hydra* (A'), echinoderms (B) *and* chordates (C). Cooption of Wnt/β-catenin signaling (violet dot), nodal signaling (yellow dot), and opposing regulatory mechanism between Nodal and Bmp signaling (green dot) for the establishment of organizing centers of embryonic axes is indicated. Mechanisms involved in left-right axial patterning are not shown. For references see text.

half of the embryo. Interestingly, β-catenin is one of the earliest dorsal signal determinants that is present in a strongly asymmetric manner in the amphioxus embryo as early as the cleavage stage. According to the proposed fate map of amphioxus embryo, the region of dorsal activity is located in the vegetal part of the blastula (*Holland and Holland, 2007*). Although at the 64-cell stage we are not able to exactly define which part of the vegetal half is presaged to become dorsal in the amphioxus embryo, colocalization of nuclear β-catenin with *Goosecoid* and P-Smad2 over the blastula and early gastrula stages allows us to presume that the vegetal blastomeres with accumulated nuclear β-catenin might represent the presumptive dorsal region in the blastula of amphioxus embryo and could be homologized with Nieuwkoop center in vertebrates. Otherwise, the fast change in the site of Wnt/β-catenin activity would have occurred in the amphioxus embryo over the short period from the cleavage to blastula stage, which is less likely. We do not know the mechanism that drives the asymmetrical accumulation of β-catenin in the early amphioxus embryo. In fact, the mechanisms driving the asymmetrical nuclear localization of β-catenin at the cleavage stages are not fully understood even in vertebrates, although the role of Wnt/β-catenin signaling in initial axis establishment is well described. In vertebrates, one of the proposed mechanisms is based on evidence that β-catenin-stabilizing components of the Wnt/β-catenin pathway are translocated during cortical rotation to the vegetal dorsal region (*Weaver and Farr, 2003*; *Miller et al., 1999*; *Houston, 2017a*). The second model suggests that enrichment of nuclear β-catenin is a result of the accumulation of Wnt signaling proteins in the dorsal blastomeres (*Tao et al., 2005*; *Lu et al., 2011*; *Cha et al., 2008*). Both these mechanisms may operate in the amphioxus. A recent study demonstrated the existence of cytoskeletal reorganization that takes place in the amphioxus embryo during the first minutes after fertilization and influences the distribution of maternal *Nodal* mRNA (*Morov et al., 2016*), indicating that cytoplasmic movements are present in the early amphioxus embryo and might play a role in the β-catenin distribution. In addition, Wnt signaling proteins are very plausible candidates for activating vegetal accumulation of nuclear β-catenin in the early amphioxus embryo. *Wnt11*, the orthologue of which was proposed to be the activator of early Wnt/β-catenin signaling in *Xenopus* (*Tao et al., 2005*), cannot be excluded as a possible candidate also in amphioxus due to its early expression in the embryo (*Qian et al., 2013*; *Somorjai et al., 2018*).

## Dual stage-dependent role of Wnt/β-catenin signaling during early development of chordates

In zebrafish and *Xenopus* embryos, a remarkable feature of Wnt/β-catenin signaling role in axial patterning is a time-dependent dramatic switch from promoting the dorsal and anterior cell fate shortly after fertilization to propagating the ventral and posterior cell fate at the gastrula stage (*Bellipanni et al., 2006*; *Ding et al., 2017*; *Zinski et al., 2018*; *Houston, 2017a*; *Sokol, 2015*). Shortly after fertilization, β-catenin is accumulated in the dorsal vegetal region of the embryo and Wnt/β-catenin signaling activates dorsal-specific target genes (*Kelly et al., 2000*; *Tao et al., 2005*; *Ding et al., 2017*; *Skirkanich et al., 2011*). However, somewhat later, at the end of blastula and during gastrulation, the Wnt/β-catenin signaling activity moves to the ventral, lateral and posterior position and its function is switched from the activator to the repressor of dorsal-specific genes (*Schohl and Fagotto, 2002*; *Bellipanni et al., 2006*; *Shimizu et al., 2012*; *Kjolby and Harland, 2017*; *Nakamura et al., 2016*). In amphioxus, the involvement of Wnt/β-catenin in promoting the posterior and suppressing the anterior/dorsal cell fate was previously reported (*Holland et al., 2005*; *Onai et al., 2009*). In the present study, the experiments demonstrating chemical treatments with an activator of Wnt/β-catenin signaling at the blastula stage are in agreement with the previous results suggesting an evolutionarily conserved role of the pathway in anterior/posterior patterning (*Petersen and Reddien, 2009*). However, up to now the role of Wnt/β-catenin signaling in the initiation of the dorsal/anterior cell fate was considered to be a vertebrate innovation (*Holland et al., 2005*; *Yasui, 2017*). A widely accepted current view is that Nodal/Vg1 is only a signaling pathway that specifies the dorsal/anterior identity in the early amphioxus embryo and evokes its neural induction (*Le Petillon et al., 2017*; *Onai et al., 2010*; *Morov et al., 2016*; *Yasui, 2017*; *Holland and Onai, 2012*). Here, we showed that, similarly as in zebrafish and *Xenopus*, the Wnt/β-catenin-dependent propagation of the dorsal cell fate operates during a limited period of early amphioxus development. In fact, even the treatment of the embryo with a low, 3 µM concentration of CHIR99021 caused conspicuous changes in the phenotype when applied at the one-cell stage, while the application of a higher, 10 µM amount of the substance after the cleavage stage affected the development

only modestly. The differing results obtained by *Holland et al., 2005* and (*Yasui et al., 2002*) after the stimulation of Wnt/β-catenin signaling could be explained by using LiCl as an activator. LiCl and CHIR99021 activate the Wnt/β-catenin signaling by inhibition of glycogen synthase kinase (GSK3), a component of the β-catenin destruction complex, but the mechanisms of function are different (*Freland and Beaulieu, 2012*; *Ring et al., 2003*). One of the suggested molecular mechanism by which LiCl inhibits GSK3 is that $Li^+$ ions act as competitive inhibitors for the binding of co-factor $Mg^{2+}$ to GSK3 (*Ryves and Harwood, 2001*). Therefore, the potency of LiCl to activate Wnt/β-catenin signaling is dependent on the $Mg^{2+}$ concentration. Artificial sea water applied in the experiments of *Holland et al., 2005* contained $Mg^{2+}$ and *Yasui et al., 2002* used natural sea water, in which the $Mg^{2+}$ ions are always present. It is probable that the difference in the $Mg^{2+}$ concentration caused the inconsistence of the results of *Yasui et al., 2002* and *Holland et al., 2005*. CHIR99021 was described as a more selective and substantially more potent inhibitor of GSK3 than LiCl (*Freland and Beaulieu, 2012*), which might be necessary for noticeable upregulation of the Wnt/β-catenin signaling during a short period of time. Interestingly, Yasuoka et al. have recently shown that the cis-regulatory sequence of amphioxus *Goosecoid* is positively regulated by Wnt/β-catenin signaling when injected into the *Xenopus* embryo (*Yasuoka et al., 2019*). The authors state that it is surprising in the context of the idea that Wnt/β-catenin signaling does not play any role in the formation of amphioxus dorsal organizer. Evidently, their observation is consistent with the present study. We additionally performed a reporter luciferase assay in human cell culture and obtained similar results (*Figure 9—figure supplement 1*). Combined, our experiments using activators and inhibitors of Wnt/β-catenin signaling suggest that the normal development of the amphioxus embryos is highly dependent on the stage of the treatment application. Obviously, similarly as in vertebrates, a switch of the role of Wnt/β-catenin signaling from the activator to the repressor of the dorsal cell fate occurs at the end of the blastula stage in cephalochordates.

Our functional data are consistent with the mapping of Wnt/β-catenin signaling activity during early development. At the early gastrula stage, nuclear β-catenin is coexpressed with the P-Smad2 and Goosecoid proteins. Thereafter, during gastrulation, nuclear β-catenin and *Axin* appear in the ventral and dorsal posterior regions, where neither Chordin nor Goosecoid are expressed. Interestingly, in *Xenopus* (*Schohl and Fagotto, 2002*) and zebrafish (*Shimizu et al., 2012*), Wnt/β-catenin signaling activity emerges in the ventral and lateral sides of the embryo at the late blastula and early gastrula stages. In contrast, in amphioxus, conspicuously higher signal of nuclear β-catenin and *Axin* is concentrated in the dorsal endomesoderm at the early gastrula stage. Thereafter, at the late gastrula and early neurula stage, the expression of nuclear β-catenin shrinks to the most posterior dorsal and ventral mesoderm. In fact, during early amphioxus development, Wnt/β-catenin signaling never appears throughout the ventral and lateral endomesoderm as is the case of vertebrates. The *Axin* and β-catenin posterior localization at the late gastrula and early neurula stage corresponds to the expression of the amphioxus *Wnt8* gene (*Yu et al., 2007*; *Schubert et al., 2000b*). In zebrafish and *Xenopus*, Wnt8 regulates patterning of the ventral mesoderm through the activation of *Vent* genes at the gastrula stage (*Hoppler and Moon, 1998*; *Ramel et al., 2005*). In contrast, amphioxus *Vent* genes are not activated by Wnt/β-catenin signaling during gastrulation (*Kozmikova et al., 2011*). It is possible that in the amphioxus, as it was previously suggested in *Onai et al., 2009* and *Holland et al., 2005*, canonical Wnt signaling is predominantly required for the propagation of the posterior cell fate after the dorsal organizer formation. As it was already discussed in *Schubert et al., 2000b*; *Kozmik et al., 2001* and *Kozmikova et al., 2011*, cooption of Wnt/β-catenin signaling in ventral mesoderm patterning might be related to the necessity of massive and complex embryonic structure to form a highly efficient circulatory system.

## Wnt/β-catenin and Nodal signaling cooperation during early dorsal cell fate establishment is conserved in chordates

In this study, we demonstrated that Nodal and Wnt/β-catenin signaling cooperate early in development to initiate the dorsal cell fate and establish the correct dorsoanterior/ventroposterior axis in the amphioxus embryo. Moreover, their cooperation is important for promoting the Nodal signaling itself. Cooperation of Wnt/β-catenin and Nodal signaling in the activation of dorsal organizer genes was also described in zebrafish (*Dougan et al., 2003*), *Xenopus* (*Reid et al., 2012*) and recently in human gastruloids obtained from human embryonic stem cells (*Martyn et al., 2018*). We have shown that, similarly as in zebrafish (*Kelly et al., 2000*; *Dougan et al., 2003*) and *Xenopus*

(*McKendry et al., 1997*; *Takahashi et al., 2000*; *Agius et al., 2000*), Wnt/β-catenin signaling is necessary for the expression of *Nodal* and *Vg1* in the dorsal organizer of the amphioxus. Both *Nodal* and *Vg1* are maternally expressed, although their expression differs (*Onai et al., 2010*). *Vg1* is ubiquitously distributed from the beginning of development and its first asymmetric expression is observed in the dorsal region of the mid-gastrula stage (*Le Petillon et al., 2017*; *Onai et al., 2010*). *Nodal* is asymmetrically expressed in the animal half of the embryo over the period from the start of development to the 64-cell stage. The enrichment of the Nodal transcripts in the presumptive dorsal vegetal region is first observed at the blastula stage (*Onai et al., 2010*; *Morov et al., 2016*). The described expression pattern of genes encoding Nodal signaling ligands is consistent with our finding indicating the first noticeable asymmetry in the distribution of nuclear P-Smad2 at the blastula stage. Asymmetric distribution of β-catenin is seen earlier than in the case of the mediator of Nodal signaling. Therefore, taking into account that Wnt/β-catenin signaling can initially activate the expression of *Nodal* and *Vg1*, it is highly probable that Wnt/β-catenin signaling is one of the key factors acting to enrich Nodal signaling activity in the dorsal region of the amphioxus embryo. Presumably, ubiquitous Nodal signaling activity, which is present at the 64-cell stage, is initiated by *Nodal* and *Vg1*, and, subsequently, is promoted by Wnt/β-catenin in the presumptive dorsal organizer region. In sea urchin, Wnt/β-catenin signaling induces expression of *Nodal* in the blastula ventral ectoderm, which is considered to be the functional equivalent of Spemann organizer (*Lapraz et al., 2015*). Thus, the Wnt/β-catenin signaling-dependent expression of *Nodal* in the organizer domain of early embryo could be already present in the common ancestor of chordates and Ambulacraria (*Figure 10*). Although in sea urchin nuclear β-catenin is not found in the *Nodal*-positive cells of the ventral ectoderm, Wnt/β-catenin signaling induces expression of Nodal from the adjacent vegetal blastomeres by a double repression mechanism involving FoxQ2 transcription factor (*Yaguchi et al., 2008*; *Duboc et al., 2004*). In amphioxus, the domains of Nodal and Wnt/β-catenin signaling activity during the organizer formation are overlapping, and thus it is possible that Wnt/β-catenin signaling directly activates *Nodal* and *Vg1* expression. To get a clue whether the regulation could be direct, we analyzed the cis-regulatory sequences of amphioxus *Nodal* and *Vg1* with luciferase reporter assay (*Figure 9—figure supplement 1*). However, we did not observe any conspicuous upregulation of the reporter gene upon stimulation of the canonical Wnt signaling. It is possible that the analyzed cis-regulatory elements did not contain functional Tcf-binding sites. Interestingly, amphioxus FoxQ2 is negatively regulated by Wnt/β-catenin signaling during the blastula and gastrula stages (*Kozmikova et al., 2011*) and its expression appears at the blastula stage (*Yu et al., 2003*), when asymmetrical distribution of Smad2 is first observed. We can speculate that if amphioxus FoxQ2, similarly as in sea urchin, is a repressor of the *Nodal* gene, then the mechanism of double repression activation of Nodal signaling could operate during the establishment of Smad2 activity in the dorsal organizer of amphioxus embryo. Furthermore, our data show that Nodal signaling in cooperation with Wnt/β-catenin signaling is necessary for the expression of *Nodal* itself in the dorsal organizer, indicating the existence of an autoregulatory loop mediating feedforward activation of the Nodal activity during early development. Autoactivation of Nodal genes operates in the amphioxus embryo at the late gastrula and early neurula stages (*Le Petillon et al., 2017*; *Li et al., 2017*) Moreover, the expression of Nodal during the early organizer formation relies on autoregulation in zebrafish (*Feldman et al., 1998*; *Dougan et al., 2003*), *Xenopus* (*Onuma et al., 2002*) and sea urchin (*Range et al., 2007*), suggesting the evolutionary conservation of this mechanism in Deuterostomes (*Figure 10*).

Interestingly, similar core-signaling cassette involving Wnt/β-catenin acting upstream of autoactivating Nodal signaling acts to induce a secondary axis formation in cnidarian *Hydra* (*Watanabe et al., 2014*; *Figure 10*). In Bilateria, Nodal signaling is involved in the induction of the secondary symmetry break during the formation of dorsal organizer (*Takahashi et al., 2000*; *Jones et al., 1995*; *Osada and Wright, 1999*; *Birsoy et al., 2006*; *Agius et al., 2000*; *Hoodless et al., 1999*; *Feldman et al., 1998*; *Niederländer et al., 2001*; *Martyn et al., 2018*; *Gritsman et al., 2000*; *Zhou et al., 1993*; *Collignon et al., 1996*; *Chea et al., 2005*) and the establishment of left-right asymmetry (*Soukup et al., 2015*; *Grande and Patel, 2009*; *Duboc et al., 2005*). So far only limited data suggest that Nodal functions downstream of Wnt/β-catenin signaling in vertebrate left-right specification (*Nakaya et al., 2005*; *Zhang et al., 2012*). Nevertheless, it is likely that during metazoan evolution Wnt/β-catenin-Nodal module was repeatedly coopted for axial patterning as discussed in *Watanabe et al., 2014*.

In sea urchin, Nodal signaling from the organizer is necessary for promoting Smad1/5/8 signaling, in the dorsal part of the blastulae embryo (*Lapraz et al., 2015*). In contrast, the opposing regulatory interactions between Nodal and Smad1/5/8 signaling underlay the dorsal and ventral cell fate specification in cephalochordate amphioxus and vertebrates (*Jones et al., 1995*; *Onai et al., 2010*; *Kozmikova et al., 2013* and reviewed in *Houston, 2017b* and *Zinski et al., 2018*; *Figure 10*). In early amphioxus embryo, the activity gradient of the Bmp signaling pathway starts to be asymmetric at the 32-cell stage and is next enriched in the presumptive ventrolateral regions of the embryo (*Kozmikova and Yu, 2017*). Thus, two pathways shape the Nodal/Smad2 activity into the dorsal domain of the embryo: Bmp signaling by ventrolateral repression and Wnt/β-catenin signaling by dorsal promotion. Together, Wnt/β-catenin, Nodal and Bmp signaling function to specify the correct dorsal and ventral cell fate in the early chordate embryo (*Figure 10*).

# Materials and methods

**Key resources table**

| Reagent type (species) or resource | Designation | Source or reference | Identifiers | Additional information |
|---|---|---|---|---|
| Strain, strain background (*Branchiostoma lanceolatum*) | Wild type | collected in Argeles-sur-mer, France | NCBITaxon: 7740 | |
| Strain, strain background (*Branchiostoma floridae*) | Wild type | Laboratory cultures, Institute of Molecular Genetics | NCBITaxon: 7739 | |
| Chemical compound, drug | CHIR99021 | SelleckChem | S1263 | |
| Chemical compound, drug | C59 | Xcess Bioscience Inc | M-60005–2 s | |
| Chemical compound, drug | SB505124 | Sigma | S4696 | |
| Antibody | Anti-β-Catenin, produced in rabbit | Sigma | C2206 | (1:500) |
| Antibody | Anti-Smad2 (phospho S255), produced in rabbit | Abcam | ab188334 | (1:20,000) |
| Antibody | Anti-β-Catenin, produced in mouse | *Bozzo et al., 2017* | | (1:200) |
| Antibody | Anti-FoxA, produced in mouse | *Bozzo et al., 2017* | | (1:200) |
| Antibody | Anti-Goosecoid, produced in mouse | This paper | | (1:200) |
| antibody | Anti-Brachyury, produced in mouse | This paper | | (1:200) |
| Antibody | Anti-Elav, produced in mouse | This paper | | (1:200) |
| Sequence-based reagent | Goosecoid_fw | This paper | | GCTTCGGATCCGAAGTCACCGAGCGCACGACC |
| Sequence-based reagent | Goosecoid_rev | This paper | | GGTCTAAGCTTACTGTCCGTCGCTAGGCGT |
| Sequence-based reagent | Brachyury_fw | This paper | | GCTGTGGATCCGGAATGGAAGATTTGCAAGAT |
| Sequence-based reagent | Brachyury_rev | This paper | | CCTGTAAGCTTAGAGCGATGGTGGAGTCAT |
| Sequence-based reagent | Elav_fw | This paper | | GTGCTGGATCCGGCTCGCCCGACGGACGCAC |

*Continued on next page*

*Continued*

| Reagent type (species) or resource | Designation | Source or reference | Identifiers | Additional information |
|---|---|---|---|---|
| Sequence-based reagent | Elav_rev | This paper | | GTGCTAAGCTTACAGTCC ACTCACGTACAAGTT |
| Sequence-based reagent | Vg1_fw | This paper | | GCGGCCACTGCTTTTACTTT |
| Sequence-based reagent | Vg1_rev | This paper | | GTGTAGTAGTGGA CAAGACTGAGGG |
| Sequence-based reagent | Nodal_fw | This paper | | TAACGTGAGGCCAGGTGATG |
| Sequence-based reagent | Nodal_rev | This paper | | CGGAGTTTGCGTGTTCGACT |
| Sequence-based reagent | Wnt8_fw | This paper | | GGAATTCGCACGAT GCTCGCACAGCTC |
| Sequence-based reagent | Wnt8_rev | This paper | | GGAATTCCTAGTTGT TTCCCCTGTTTCTTC |
| Sequence-based reagent | Wnt11_fw | This paper | | CTCTTCTTCAACCTGCGACTG |
| Sequence-based reagent | Wnt11_rev | This paper | | TTTTCTCTGGCTTTCCCTTGA |
| Sequence-based reagent | dnTcf_fw | This paper | | GCGATCTAGAGGTTCCA AGCCACCGTTGCAA |
| Sequence-based reagent | dnTcf_rev | This paper | | GCTCAAGCTTTCACGT GTTGGGCGTTGACTT |
| Software, algorithm | Fiji | *Schindelin et al., 2012* | | |
| Software, algorithm | Leica Application Suite X (LAS X) | Leica | | |

## Animal collection and spawning induction

Adults of *Branchiostoma lanceolatum* were collected in Argeles-sur-mer (France) and transported to the Institute of Molecular Genetics (Prague, Czech Republic). The animals were maintained in the lab and the spawning was induced as described in *Yong et al., 1891*; *Fuentes et al., 2007*. Adults of *Branchostoma floridae* were originally collected in Tampa (Florida) and subsequently maintained as a continuous breeding culture in the laboratory. For raising and maintenance, larvae and adults are provided with brown phytoplankton (Isochrysis) originally obtained from NCMA at the Bigelow Laboratory for Ocean Sciences (East Boothbay, ME, USA). Animals are raised at 25°C and adults are kept on a 10 hr dark/14 hr light cycle. Ripe animals are selected and maintained at 14°C for at least 6 weeks. Then the animals are moved to 28°C for 24 hr. When the light is turned off, 20–80% of the animals usually spawn 1 hr later.

## Pharmacological treatments and mRNA injection

Collected eggs from *Branchiostoma lanceolatum* were fertilized and the embryos were transferred to 6 cm Petri dishes. The embryos were kept at 19 °C and treated for 15 min after fertilization at one-cell stage or 4.5 hr after fertilization at the blastula stage with different chemicals. The blastula stage was defined according to *Hirakow and Kajita, 1990*. Control embryos were treated with dimethyl sulfoxide (DMSO). C59 was applied to inhibit and CHIR99021 to activate the Wnt/β-catenin signaling pathway (*Supplementary file 1*). To inhibit the Nodal signaling pathway, the embryos were treated with SB505124. The applied concentrations were determined in preliminary analysis using following concentration ranges of drugs for each chemical: C59: 0.3 μM, 1 μM, 3 μM and 10 μM; CHIR99021: 3 μM, 5 μM, 10 μM and 20 μM; SB500124 0.5 μM, 1 μM, 10 μM and 30 μM. After treatment, the embryos were fixed in 4% PFA/MOPS buffer overnight for in-situ hybridization or 15 min on ice for immunostaining as described in *Yong et al., 1891*.

The coding regions of amphioxus Wnt8, Wnt11 and dnTcf were subcloned into pCS2+ using primers Wnt8_fw/Wnt8_rev, Wnt11_fw/Wnt11_rev and dnTcf_fw/dnTcf_rev. RNA was prepared using

the mMessage mMachine kit (Ambion) and purified by RNeasy Micro kit (Qiagen). mRNA injection was performed as previously described (*Kozmikova et al., 2013*). The injected bubble occupied one tenth of the volume of the unfertilized egg. The concentrations of injected Wnt8 mRNAs were 3 ng/µl (low dose), 7 ng/µl (high dose), and 21 ng/µl. The embryos injected with *Wnt8* mRNA at a concentration of 21 ng/µl were not able to undergo gastrulation properly and died during the gastrulation. The concentrations of injected Wnt11 mRNA were 7 ng/µl, 20 ng/µl, 50 ng/µl (low dose) and 250 ng/µl (high dose). Microinjection of *Wnt11* mRNA at a concentration of 7 ng/µl and 20 ng/µl had no effect on the normal development of amphioxus embryos. The concentration of injected dnTcf mRNA was 500 ng/µl.

## In situ hybridization and immunostaining

For in situ hybridization, the fixed embryos were transferred to 70% ethanol and stored for at least several days at −70℃. In situ hybridization was performed as previously described (*Kozmikova et al., 2013*) and carried out at least three times with 20 embryos each time for each gene and experimental conditions. For immunostaining, the embryos after fixation were immediately transferred to 100% methanol or 70% ethanol. The embryos were staging according to *Hirakow and Kajita, 1990*, *Hirakow and Kajita, 1991*, and *Hirakow and Kajita, 1994*. Immunostaining was generally performed as described in *Bozzo et al., 2017* with some modifications: all washing solutions contained 1xTBS buffer with 0.05% of Triton. Blocking solution contained 8% BSA and 10% donkey serum, and donkey anti-mouse or anti-rabbit secondary antibody was used. Each individual staining of protein was performed with at least 20 embryos repeatedly at least three times. For confocal microscopy, the samples were mounted in VECTASHIELD (Vector Laboratories, Inc) to glass depression slides for mid-gastrula, neurula and larval stages or glass bottom dishes (MatTek) for embryos at the cleavage, onset of gastrulation and early gastrula stages. The series of Z-stack imaging were taken with a Leica SP8 confocal microscope and processed with FIJI image analysis software for three-dimensional reconstruction.

## Generation of antibodies recognizing amphioxus proteins

Partial coding regions of amphioxus Elav, Goosecoid and Brachyury (Bra1) were subcloned into pET42a using primers Elav_fw/Elav_rev, Goosecoid_fw/Goosecoid_rev and Brachyury_fw/Brachyury_rev. Expression and purification of proteins for immunization were performed as described in *Bozzo et al., 2017*. Mice of the B10A-H2xBALB/CJ strain were immunized four times in 3 weeks interval. Housing of mice and in vivo experiments were performed in compliance with the European Communities Council Directive of 24 November 1986 (86/609/EEC) and national and institutional guidelines. Animal care and experimental procedures were approved by the Animal Care Committee of the Institute of Molecular Genetics (no. 71/2014). Specificity of sera were validated by correlation between whole-mount in situ hybridization pattern of a given gene with that obtained by whole-mount immunohistochemistry.

## Cell culture, transient transfection and luciferase reporter assay

Cell culture, transient transfection and luciferase reporter assay were performed as described in *Kozmikova et al., 2011*. 293 T cells were purchased from ATCC and authentication of the cell line was confirmed by STR profiling. Cell line was tested negative for mycoplasma contamination by Mycoplasma Detection Kit (Lonza). Nodal and Vg1 promoters were amplified from *B. lanceolatum* genomic DNA using primers Nodal_fw/Nodal_rev and Vg1_fw/Vg1_rev and subcloned into pGL3-basic vector. AmphiGoosecoid-luciferase reporter gene and expression vector encoding stabilized β-catenin were described previously (*Kozmikova et al., 2011*).

## Acknowledgements

This work was supported by GACR 15–21285J (awarded to IK), GACR 17–15374S (awarded to ZK), European Regional Development Fund-Project No. CZ.02.1.01/0.0/0.0/16_013/0001775, and RVO68378050-KAV-NPUI. We thank the Microscopy Centre – Light Microscopy Core Facility, IMG ASCR, for support with the confocal microscopy presented herein and Sarka Takacova for proofreading the manuscript.

## Additional information

### Funding

| Funder | Grant reference number | Author |
|---|---|---|
| Grantová Agentura České Republiky | GACR 15-21285J | Iryna Kozmikova |
| Grantová Agentura České Republiky | GACR 17-15374S | Zbynek Kozmik |
| Ministerstvo Školství, Mládeže a Tělovýchovy | ERDF project No. CZ.02.1.01/0.0/0.0/16_013/0001775 | Iryna Kozmikova |

The funders had no role in study design, data collection and interpretation, or the decision to submit the work for publication.

### Author contributions

Iryna Kozmikova, Conceptualization, Data curation, Formal analysis, Funding acquisition, Investigation, Methodology, Writing - original draft, Project administration, Writing - review and editing; Zbynek Kozmik, Resources, Data curation, Funding acquisition, Investigation, Methodology, Writing - review and editing

### Author ORCIDs

Iryna Kozmikova (iD) https://orcid.org/0000-0002-7861-9802

### Decision letter and Author response

Decision letter https://doi.org/10.7554/eLife.56817.sa1
Author response https://doi.org/10.7554/eLife.56817.sa2

## Additional files

### Supplementary files

• Supplementary file 1. Chemicals used in this study to affect the signaling pathways.

• Transparent reporting form

### Data availability

All data generated or analysed during this study are included in the manuscript and supporting files.

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
