## [Decision Letter]

**Acceptance summary:**

This is a very interesting manuscript on the role of Wnt/β-Catenin signaling in Amphioxus (cephalochordates), a small group of basal chordates which are of crucial importance for our understanding of deuterostome evolution. While the expression and function of Wnt signaling during embryogenesis of these animals has been described previously, there was no conclusive picture of the role of Wnt/β-Catenin signaling in axial patterning and organizer formation. In the present paper, the authors reanalyzed Wnt/β-catenin signaling using an Amphioxus-specific β-Catenin antibody and loss-of-function / gain-of-function experiments. They performed a thorough and compelling analysis that shows exciting similarities in axis formation between cephalochordates and vertebrates.

**Decision letter after peer review:**

Thank you for submitting your article "Wnt/β-catenin signaling is an evolutionarily conserved determinant of chordate dorsal organizer" for consideration by *eLife*. Your article has been reviewed by three peer reviewers, and the evaluation has been overseen by a Reviewing Editor and Marianne Bronner as the Senior Editor. The following individuals involved in review of your submission have agreed to reveal their identity: Thomas Holstein (Reviewer #2); Christof Niehrs (Reviewer #3).

Summary:

The reviewers have discussed the reviews with one another and the Reviewing Editor has drafted this decision to help you prepare a revised submission. The reviewers agree that you have generated new insight into the role of Wnt/ß-catenin signaling during the embryonic development of two species of amphioxus. Your main conclusion is that Wnt/ß-catenin is necessary for the formation of the dorsal organizer and acts in concert with Nodal signaling, as it occurs in vertebrates. The result is viewed as interesting for the field as it contradicts results from a previous study and thereby suggests that the last common ancestor of all chordates already used the strategy of Wnt/ß-catenin activity or the formation of the dorsal organizer.

Essential revisions:

Each of the reviewers has made suggestions to improve the manuscript. I attach the reviews below. It seems to me that none of the comments require new experimentation. I am looking forward to receiving an updated and improved version of the work.

Reviewer #1:

In this study, the authors studied the role of Wnt/ß-catenin signaling during the embryonic development of two species of amphioxus. The main conclusion of their work is that as in vertebrates, Wnt/ß-catenin is necessary for the formation of the dorsal organizer and acts in concert with Nodal signaling. This result is of interest as it contradicts a previous low-quality study on the topic and thereby strongly suggests that the last common ancestor of all chordates already used this strategy. The scope of the work is thus adapted to the broad readership of *eLife*.

As it stands, however, the work does not reach the quality standard expected from an *eLife* article. Some inconclusive experiments are included and some claims are insufficiently substantiated. The work needs some formal, and perhaps some experimental work before it reaches these standards.

Main comments:

The authors do not show evidence that the antibodies used are specific for the proteins of interest. Some of these antibodies were previously described in Bozzo et al., 2017, but this publication does not show either that the antibodies specifically recognize a single protein, using Western blot for instance.

The phenotypes obtained with JW74 are not very convincing, and this compound is less specific for Wnt signaling (it blocks tankyrases) than C59 or the overexpression of dnTcf. I am not sure adding experiments with this compound is necessary, but if the authors think they are, the stages analyzed with C59 and JW74 should be identical. What happens when C59 is applied during blastula stages? Does this still lead to a failure to form axial structures? If not, this suggests that the target Wnt gene is of maternal origin.

I would have expected CHIR treatment from the zygote stage would lead to the complete vegetalization of the embryos. Is this observed (is ß-catenin found nuclear throughout the embryo?) and could this explain the gastrulation phenotype observed by the authors? I am aware that making new experiments in this Covid period is difficult/impossible but the authors should at least comment on these issues. Figure 6 suggests that there is no major increase in endoderm as documented by the presence of both Brachyury and Elav expression. Overall, the analysis at the larval stage is made difficult by the very disturbed morphogenesis induced by the inhibitor. Finally, the analysis of *Chordin* expression at the gastrula stage (Figure 5A j-k') is not very conclusive. It could be interpreted as overall background staining and I am not sure whether the gene is up- or down-regulated.

Molecular phenotypes obtained following Wnt8 overexpression are very difficult to interpret. I do not think they add to the story. Overexpression of Wnt11 is more interesting, but the pattern of expression of this gene in the organizer suggests that inhibition of Wnt synthesis at this stage should lead to the repression of axis formation (see comment above on the timing of action of C59). Does Wnt11 mRNA microinjection lead to early ß-catenin nuclear localization? If it does, and this gene is in reality expressed much later, then it is a poor candidate for the endogenous Wnt starting the cascade and its overexpression does not add to the experiments carried out with CHIR. It could at best be mentioned briefly in a supp figure.

Overall the manuscript is at places very technical. It would gain from a substantial shortening, removing the pieces of data that are least conclusive. It would also benefit from some reorganization, which could involve grouping together the various treatments that repress Wnt signaling (C59, JW74 – if this compound is kept in the final version-, dnTcf) on the one hand and those that activate this pathway (CHIR treatment, Wnts overexpression) on the other.

Reviewer #2:

This is a very interesting manuscript on the role of Wnt/β-Catenin signaling in Amphioxus (cephalochordates), a small group of basal chordates which are of crucial importance for our understanding of deuterostome evolution. While in several papers the expression and function of Wnt signaling during embryogenesis of these animals has been described, there is no conclusive picture of the role of Wnt/β-Catenin signaling in axial patterning and organizer formation so far. In the manuscript under review, the authors reanalyzed Wnt/β-catenin signaling. Using an Amphioxus-specific β-Catenin antibody and loss-of-function / gain-of-function experiments they performed a thorough and compelling analysis that shows exciting similarities in axis formation with vertebrates. They found in embryos up to early gastrulation an asymmetrical distribution of nuclear β-Catenin, which overlaps with that of goosecoid expression, an organizer-specific gene, and P-Smad2, the crucial downstream factor of activated Nodal signaling. This suggests that Wnt/β-Catenin might have a similar function in setting up the dorsal organizer (with Nodal as a crucial component) as in vertebrates. Their main finding is supported by a number of functional experiments and specific marker genes for organizer formation (chordin, goosecoid, elav, neurogenin, nodal, vg1), but also by microinjection experiments using wnt8/wnt11 mRNA that induced ectopic dorsal axes in Amphioxus larvae. A deeper functional analysis of the interactions between Wnt/β-catenin signaling and Nodal signaling clearly requires a genetic analysis of both pathway in Amphioxus organizer formation. However, the pharmacological experiments (C59 inhibitor and CHIR99021 activator of Wnt/β-catenin signaling, SB505124 inhibitor of Nodal signaling) presented in the current manuscript already indicate a cooperativity of Nodal and Wnt/β-Catenin signaling in establishing the organizer. This is already a significant and major step forward in our understanding of the origin of the vertebrate organizer.

There are also some weaknesses in the current manuscript the authors should address in a revised version. The interaction of Nodal and Wnt/β-Catenin signaling is essential for the chordate organizer, but it is not restricted to deuterostomes as shown in Figure 10. Among others Grande and Patel, 2009, and Watanabe et al., 2014, reported a function of Nodal signaling downstream of Wnt/β-Catenin signaling in mollusks and cnidarians. While Nodal signaling might have been differentially co-opted in deuterostomes and protostomes during animal evolution its occurrence and relevance should be discussed. I'm also wondering why the authors did not mention the fact that a core-signaling cassette consisting of β-Catenin, Nodal and Pitx predated the cnidarian-bilaterian split (Watanabe et al., 2014) into Figure 10. Also, the paper of Chea et al., 2005, entitled “Nodal Signaling and the Evolution of Deuterostome Gastrulation” is missing.

Reviewer #3:

In vertebrate development, Wnt signaling has an early role in specification of dorso-anterior structures and a later one in antero-posterior patterning after MBT and during gastrulation/neural patterning. While the late role is conserved in Amphioxus, previous studies suggested that the early role was not, the evidence relying on b-cat immunostaining and LiCl treatments. In this study, Kozmikova and Kozmik revise this notion and demonstrate that the role of Wnt/b-catenin signaling in early dorso-ventral axis formation is indeed conserved in Amphioxus.

While this study is overdue, it may be questioned whether the confirmation of an early role of Wnt signaling in Amphioxus in this day and age is of sufficient novelty/conceptual advance for *eLife*, or if it is better suited for a specialized journal, where most of these types of studies are typically found. Given how comprehensive the study is, its high technical quality, and the fact that it also nicely supports the conserved epistasis Wnt->nodal->goosecoid, I would support the manuscript being adequate for *eLife*.

1) Figure 2 Abiii shows asymmetric bcat on the dorsal side; Previously the authors (Bozzo et al., 2017) showed homogenous vegetal distribution of nuclear b-catenin with the same Ab at late blastula stage (Figure 3). This discrepancy is not explained.

2) Figure 6A: doubling of bra expression ("two longitudinal stripes") is only moderately convincing

3) Figure 6B: shows anterior development inhibited by late wnt activation, consistent with previous findings and evolutionary conservation that late Wnt controls a-p patterning; needs to be commented on in Results and Discussion;

4) The endogenous, maternal wnt, which controls the early dv signal in Amphioxus remains unexplored. The mRNA injection experiments with Wnt8 and Wnt11 are not informative in this regard. But identifying the endogenous Wnt doing the job may not be essential for the overall conclusion of the study.

5) Figure 7Ac axis duplication is striking, but they should indicate the "n"

6) The quote "However, overactivation of Wnt/β-catenin signaling at the mid-blastula stage does not influence the axial patterning during the amphioxus development" is misleading, since Wnt regulates amphioxus ap axial patterning, and be qualified: "However, overactivation of Wnt/β-catenin signaling at the mid-blastula stage does not influence the DORSO-VENTRAL axial patterning during the amphioxus development".

---

## [Author Response]

Reviewer #1:In this study, the authors studied the role of Wnt/ß-catenin signaling during the embryonic development of two species of amphioxus. The main conclusion of their work is that as in vertebrates, Wnt/ß-catenin is necessary for the formation of the dorsal organizer and acts in concert with Nodal signaling. This result is of interest as it contradicts a previous low-quality study on the topic and thereby strongly suggests that the last common ancestor of all chordates already used this strategy. The scope of the work is thus adapted to the broad readership of eLife.As it stands, however, the work does not reach the quality standard expected from an eLife article. Some inconclusive experiments are included and some claims are insufficiently substantiated. The work needs some formal, and perhaps some experimental work before it reaches these standards.Main comments:The authors do not show evidence that the antibodies used are specific for the proteins of interest. Some of these antibodies were previously described in Bozzo et al., 2017, but this publication does not show either that the antibodies specifically recognize a single protein, using Western blot for instance.

We understand the concerns of this reviewer, however the determination of antibody specificity is generally not a trivial task and very much depends on the application for which the antibody is used. From this point of view we appreciate the suggestion to use the Western blot with membrane-bound samples but we are certain that this will not resolve the issue of antibody specificity when applied to the whole-mount immunohistochemistry technique on PFA-fixed embryos. That is why we believe that the correlation between whole-mount in situ hybridization pattern with that obtained by wholemount immunohistochemistry represents a more appropriate (and valid) indication of antibody specificity (see Author response image 1). We provided the explanatory text in Materials and methods section dealing with antibody generation.

**Author response image 1. sa2fig1:** 

We would like to point out, however, that in the current study we use some of the antibodies to merely label the domains/structures of the developing amphioxus embryo such as dorsal blastopore lip (Goosecoid antibody), the notochord (Brachyury antibody), or the neural tube (Elav antibody). Although in our view the expression as determined by in situ hybridization and nuclear localization of the immunohistochemistry signal in case of transcription factors is a strong indication that the antibodies used in the current study specifically detect their presumed antigen, for the purpose of labelling embryonic structures even antibodies recognizing unknown epitopes would serve the purpose (if such antibodies existed). In case of the key ß-catenin antibody, we have shown in Bozzo et al., 2017, that there is an identical pattern using an independent antibody produced in different host species, making it highly unlikely that the antibody detects another amphioxus protein (other than ß-catenin).

The phenotypes obtained with JW74 are not very convincing, and this compound is less specific for Wnt signaling (it blocks tankyrases) than C59 or the overexpression of dnTcf. I am not sure adding experiments with this compound is necessary, but if the authors think they are, the stages analyzed with C59 and JW74 should be identical. What happens when C59 is applied during blastula stages? Does this still lead to a failure to form axial structures? If not, this suggests that the target Wnt gene is of maternal origin.

Given the inherent difficulty in obtaining amphioxus embryos and seasonality of spawnings this project lasted several years, with JW74 being used in initial experiments as soon as the compound became available (via the authors of the original study). We agree that JW74 is not only less potent in inhibiting Wnt/ß-catenin signalling but also less specific compared to C59, and the data represent the weak part of the manuscript. Therefore, we decided to follow the suggestion of this reviewer and we removed the data using JW74 inhibitor from our revised manuscript.

We believe that, while pursuing additional questions raised by this reviewer would be of interest for potential future work, experiments involved would certainly require extensive time period if done properly on seasonal amphioxus, and are beyond the focus of the current manuscript. We would like to point out that our primary aim in the current study was to find out if Wnt/ß-catenin signalling plays any role in the establishment of the dorsal organizer and DV axis formation. At the molecular level an organizer has to be set up at the end of blastula stage/onset of gastrulation. Previous data (Onai et al., 2009) and data in our study using CHIR99021 demonstrated that late activation of Wnt/ß-catenin signalling does not disturb DV patterning. In fact, we tried C59 treatment at 4.5h blastula stage (which is about 30min prior to morphologically recognizable start of gastrulation) and we did not observe disturbance of DV patterning. Since the issue of zygotic genome activation in amphioxus is not settled it is currently impossible to claim that blastula stage treatment would discriminate between maternal and zygotic components.

I would have expected CHIR treatment from the zygote stage would lead to the complete vegetalization of the embryos. Is this observed (is ß-catenin found nuclear throughout the embryo?) and could this explain the gastrulation phenotype observed by the authors? I am aware that making new experiments in this Covid period is difficult/impossible but the authors should at least comment on these issues. Figure 6 suggests that there is no major increase in endoderm as documented by the presence of both Brachyury and Elav expression. Overall, the analysis at the larval stage is made difficult by the very disturbed morphogenesis induced by the inhibitor. Finally, the analysis of Chordin expression at the gastrula stage (Figure 5A j-k') is not very conclusive. It could be interpreted as overall background staining and I am not sure whether the gene is up- or down-regulated.

“Although the embryos treated with high concentration of CHIR99021 at 1-cell stage did not invaginate properly the blastopore ring was formed and partial invagination of vegetal plate took place. We did not observe the formation of exogastrula and strong vegetalization phenotype as described previously for sea urchin (Wikramanayake et al., 1998).”

The above paragraph was included in the revised text.

Based on the expression of *Chordin* and *Nodal* we think that the above treatment leads to transformation into dorsal cell fate. In perturbation experiments we did not stain ß-catenin but we rather used axin expression as an indication of Wnt/ß-catenin signalling activity.

Regarding *Chordin* expression at the gastrula stage we did not claim that the gene is up- or down-regulated but we stated “the expression of *Chordin* and *Nodal* spread out over the whole embryo”. We are sure that the uniformly spread signal upon CHIR99201 does not represent a background since in both untreated and treated embryos the staining was performed in parallel for the same period of time. To mitigate concerns of this reviewer we provide in Author response image 2 images of two additional independent experiments.

Molecular phenotypes obtained following Wnt8 overexpression are very difficult to interpret. I do not think they add to the story. Overexpression of Wnt11 is more interesting, but the pattern of expression of this gene in the organizer suggests that inhibition of Wnt synthesis at this stage should lead to the repression of axis formation (see comment above on the timing of action of C59). Does Wnt11 mRNA microinjection lead to early ß-catenin nuclear localization? If it does, and this gene is in reality expressed much later, then it is a poor candidate for the endogenous Wnt starting the cascade and its overexpression does not add to the experiments carried out with CHIR. It could at best be mentioned briefly in a supp figure.

We agree that mRNA injection experiments with Wnt8 and Wnt11 are not informative in elucidating the question which of the endogenous Wnt controls the early DV signal in amphioxus. We used Wnt8 and Wnt11 primarily as tools, although only Wnt11 could possibly be a candidate molecule as correctly pointed out by the reviewer. On the other hand Wnt8 mRNA injection was used in vertebrates for induction of ectopic axis. This fact was mentioned for Wnt8 case in the original version of the manuscript by the following text: “Although *Wnt8* is not expressed in the organizer of vertebrates and amphioxus, injected Wnt8 mRNA promotes ectopic axis formation in the vertebrates”. In the Discussion of the original manuscript we wrote “Wnt11 was detected by RT-PCR at the one-cell stage embryo and expressed asymmetrically in the vegetal half of the amphioxus blastula (Somorjai et al., 2018) and dorsally at the onset of gastrulation and mid-gastrula stage (Schubert et al., 2000).” By a mistake, within the Results section we only mentioned Wnt11 expression in the gastrula thus creating a confusion. We modified the text in the Results section.

In summary, we share the view with reviewer #3 that identifying the endogenous amphioxus Wnt involved in the organizer formation is not essential for the overall conclusion of the study.

Overall the manuscript is at places very technical. It would gain from a substantial shortening, removing the pieces of data that are least conclusive. It would also benefit from some reorganization, which could involve grouping together the various treatments that repress Wnt signaling (C59, JW74 – if this compound is kept in the final version-, dnTcf) on the one hand and those that activate this pathway (CHIR treatment, Wnts overexpression) on the other.

As described above we removed the JW74 data to help shortening the manuscript. We thank the reviewer for suggestion regarding manuscript reorganization and agree that it is better to group together the various treatments that repress Wnt signaling on the one hand and those that activate this pathway on the other.

Reviewer #2:This is a very interesting manuscript on the role of Wnt/β-Catenin signaling in Amphioxus (cephalochordates), a small group of basal chordates which are of crucial importance for our understanding of deuterostome evolution. While in several papers the expression and function of Wnt signaling during embryogenesis of these animals has been described, there is no conclusive picture of the role of Wnt/β-Catenin signaling in axial patterning and organizer formation so far. In the manuscript under review, the authors reanalyzed Wnt/β-catenin signaling. Using an Amphioxus-specific β-Catenin antibody and loss-of-function / gain-of-function experiments they performed a thorough and compelling analysis that shows exciting similarities in axis formation with vertebrates. They found in embryos up to early gastrulation an asymmetrical distribution of nuclear β-Catenin, which overlaps with that of goosecoid expression, an organizer-specific gene, and P-Smad2, the crucial downstream factor of activated Nodal signaling. This suggests that Wnt/β-Catenin might have a similar function in setting up the dorsal organizer (with Nodal as a crucial component) as in vertebrates. Their main finding is supported by a number of functional experiments and specific marker genes for organizer formation (chordin, goosecoid, elav, neurogenin, nodal, vg1), but also by microinjection experiments using wnt8/wnt11 mRNA that induced ectopic dorsal axes in Amphioxus larvae. A deeper functional analysis of the interactions between Wnt/β-catenin signaling and Nodal signaling clearly requires a genetic analysis of both pathway in Amphioxus organizer formation. However, the pharmacological experiments (C59 inhibitor and CHIR99021 activator of Wnt/β-catenin signaling, SB505124 inhibitor of Nodal signaling) presented in the current manuscript already indicate a cooperativity of Nodal and Wnt/β-Catenin signaling in establishing the organizer. This is already a significant and major step forward in our understanding of the origin of the vertebrate organizer.There are also some weaknesses in the current manuscript the authors should address in a revised version. The interaction of Nodal and Wnt/β-Catenin signaling is essential for the chordate organizer, but it is not restricted to deuterostomes as shown in Figure 10. Among others Grande and Patel, 2009, and Watanabe et al., 2014, reported a function of Nodal signaling downstream of Wnt/β-Catenin signaling in mollusks and cnidarians. While Nodal signaling might have been differentially co-opted in deuterostomes and protostomes during animal evolution its occurrence and relevance should be discussed. I'm also wondering why the authors did not mention the fact that a core-signaling cassette consisting of β-Catenin, Nodal and Pitx predated the cnidarian-bilaterian split (Watanabe et al., 2014) into Figure 10.

We welcome these comments a lot. We are aware of the studies of Hobmayer (Hobmayer et al., 2000) and Watanabe (Watanabe et al., 2014) in Hydra. However, we were afraid that we would be criticized for stretching the possible conservation of dorsal organizer of chordate embryos with head or budding zone organizer regions in Hydra polyp. Encouraged by the reviewer we were happy to modify the Discussion and Figure 10 to include *Hydra* data for Wnt/β-catenin-Nodal cassette. The fact that Nodal signaling might have been differentially co-opted in deuterostomes and protostomes during animal evolution is discussed in the revised manuscript. Although discussed in the text, in Figure 10 we would very much like to maintain the focus on D/V patterning in deuterostomes leaving L/R axis out.

Also, the paper of Chea et al., 2005, entitled “Nodal Signaling and the Evolution of Deuterostome Gastrulation” is missing.

We thank reviewer for pointing this out. Reference was added into Introduction and Discussion sections.

Reviewer #3:In vertebrate development, Wnt signaling has an early role in specification of dorso-anterior structures and a later one in antero-posterior patterning after MBT and during gastrulation/neural patterning. While the late role is conserved in Amphioxus, previous studies suggested that the early role was not, the evidence relying on b-cat immunostaining and LiCl treatments. In this study, Kozmikova and Kozmik revise this notion and demonstrate that the role of Wnt/b-catenin signaling in early dorso-ventral axis formation is indeed conserved in Amphioxus.While this study is overdue, it may be questioned whether the confirmation of an early role of Wnt signaling in Amphioxus in this day and age is of sufficient novelty/conceptual advance for eLife, or if it is better suited for a specialized journal, where most of these types of studies are typically found. Given how comprehensive the study is, its high technical quality, and the fact that it also nicely supports the conserved epistasis Wnt->nodal->goosecoid, I would support the manuscript being adequate for eLife.1) Figure 2 Abiii shows asymmetric bcat on the dorsal side; Previously the authors (Bozzo et al., 2017) showed homogenous vegetal distribution of nuclear b-catenin with the same Ab at late blastula stage (Figure 3). This discrepancy is not explained.

In Bozzo et al. (Bozzo et al., 2017) we did not analyse 3D whole mounts of the embryos and the figure presented there shows a single optical section (single confocal Zstack). Here we provide for the convenience of the reviewer several optical sections of the embryo 3D whole-mount of which is shown in Author response image 3.

**Author response image 3. sa2fig3:** 

2) Figure 6A: doubling of bra expression ("two longitudinal stripes") is only moderately convincing

The description “two longitudinal stripes” related to Figure 6A was deleted.

3) Figure 6B: shows anterior development inhibited by late wnt activation, consistent with previous findings and evolutionary conservation that late Wnt controls a-p patterning; needs to be commented on in Results and Discussion;

Commented in Results and Discussion.

4) The endogenous, maternal wnt, which controls the early dv signal in Amphioxus remains unexplored. The mRNA injection experiments with Wnt8 and Wnt11 are not informative in this regard. But identifying the endogenous Wnt doing the job may not be essential for the overall conclusion of the study.

We agree that our study provided limited insight into the actual endogenous Wnt molecule(s) controlling DV patterning in amphioxus. Given the inherent methodological limitations associated with amphioxus as a model the identification of such molecule(s) might take many years. We share the view that its identity is not essential for the overall conclusion of our study.

5) Figure 7Ac axis duplication is striking, but they should indicate the "n"

The "n" stating the numbers of embryos with phenotypes shown in Figure 7 are indicated in the top panel row in the bottom right corner.

6) The quote "However, overactivation of Wnt/β-catenin signaling at the mid-blastula stage does not influence the axial patterning during the amphioxus development" is misleading, since Wnt regulates amphioxus ap axial patterning, and be qualified: "However, overactivation of Wnt/β-catenin signaling at the mid-blastula stage does not influence the DORSO-VENTRAL axial patterning during the amphioxus development".

Our mistake. The sentence was corrected.